# A High-Resolution Dataset of Water Fluxes and States for Germany Accounting for Parametric Uncertainty

Matthias Zink[1], Rohini Kumar[1], Matthias Cuntz[1,2], and Luis Samaniego[1]

[1]Helmholtz Centre for Environmental Research - UFZ, Department Computational Hydrosystems, Permoserstrasse 15, 04318 Leipzig, Germany

[2]INRA, Université de Lorraine, UMR1137 Ecologie et Ecophysiologie Forestières, Champenoux, France

*Correspondence to:* Matthias Zink (matthias.zink@ufz.de)

**Abstract.** Long term, high-resolution data about hydrologic fluxes and states are needed for many hydrological applications. Because continuous large-scale observations of such variables are not feasible, hydrologic or land surface models are applied to derive them. This study aims to analyze and provide a consistent high-resolution dataset of land surface variables over Germany, accounting for uncertainties caused by equifinal model parameters. The mesoscale Hydrological Model (mHM) is employed to derive an ensemble (100 members) of evapotranspiration, groundwater recharge, soil moisture and runoff generated at high spatial and temporal resolutions (4 km and daily, respectively) for the period 1951-2010. The model is cross-evaluated against the observed daily streamflow in 222 basins, which are not used for model calibration. The mean (standard deviation) of the ensemble median Nash-Sutcliffe Efficiency estimated for these basins is 0.68 (0.09) for daily streamflow simulations. The modeled evapotranspiration and soil moisture reasonably represent the observations from eddy covariance stations. Our analysis indicates the lowest parametric uncertainty for evapotranspiration, and the largest is observed for groundwater recharge. The uncertainty of the hydrologic variables varies over the course of a year, with the exception of evapotranspiration, which remains almost constant. This study emphasizes the role of accounting for the parametric uncertainty in model-derived hydrological datasets.

# 1 Introduction

Consistent, long-term data of meteorological and hydrological variables at a high spatial resolution are needed for many applications, including i) impact assessment studies, such as for drought, flood or climate change analysis (Sheffield and Wood, 2007; Huang et al., 2010; Samaniego et al., 2013; Kumar et al., 2016; Zink et al., 2016), and ii) studies that need spatially and temporally continuous, observation-based datasets, e.g., for downscaling or disaggregating climate model outputs (Wood et al., 2004; Thober et al., 2014) or for establishing Ensemble Streamflow Prediction (Day, 1985) and reverse Ensemble Streamflow Prediction approaches (Wood and Lettenmaier, 2008).

Continuous observations of hydrologic fluxes and states are economically and logistically not feasible on regional to national scales (Vereecken et al., 2008). In-situ soil moisture observations, for example, are scarcely available. These point-scale observations are representative for a small control volume of a few cm$^3$. Evapotranspiration measurements at eddy covariance stations have footprints of tens to hundreds of meters but they are available at less than 1000 stations worldwide (FLUXNET (2007)).

Alternatives include remote sensing or reanalysis products such as NCEP-CFSR (Saha et al., 2010) or ERA-INTERIM (Dee et al., 2011). Hydrologic products derived from remote sensing are broadly available, but they do not consider the conservation of mass, i.e., the closure of the water balance. Moreover, these products are not spatially and temporally continuous due to reliance on cloud-free conditions (Mu et al., 2007; Liu et al., 2012). Reanalysis products, in contrast, provide continuous data but they have coarse spatial resolutions of at most 1/4° (Dee et al., 2016), which is not suitable for regional scale applications.

Hydrologic models driven by ground-based meteorological observations are the prime alternative to derive spatially and temporally consistent water fluxes and states at large spatial domains. For example, Maurer et al. (2002); Zhu and Lettenmaier (2007); Livneh et al. (2013); and Zhang et al. (2014) provided model-based datasets on a national scale. These data are based on the Variable Infiltration Capacity (VIC) model (Liang et al., 1994) and have, at most, a spatial resolution of 1/16° and cover the contiguous United States, Mexico, China, and parts of Canada. Livneh et al. (2015); Newman et al. (2015a); and Newman et al. (2015b) provide data on the same domain with a focus on meteorological data. A set of four models was used in the NLDAS project to assess the water balance components over the contiguous United States (Mitchell, 2004; Xia et al., 2012b, a). Studies by Nijssen et al. (2001); Fan and van den Dool (2004); Berg et al. (2005); and Sheffield et al. (2006) focus on the global domain. The spatial resolution of these global data sets is at most 1/2°, and many of these studies focus on meteorological forcings rather than hydrologic variables.

The resolution of the above mentioned model-derived datasets are coarse according to Wood et al. (2011), who stated a need for higher-resolution data and models for purposes like flood and drought forecasting. Moreover, Bierkens et al. (2015) state that water resources or river basin managers will favor highly resolved data at resolutions of 1-5 km.

The application of observational derived model products, however, also has some limitations. First, due to a limited amount of observed variables modeling approaches, like the estimation of potential evapotranspiration (PET), have to be adopted to the available data. In consequence temperature based PET methods may be preferred to more physically based approaches (e.g., radiation based). Second, the interpolation of point observations induces uncertainties depending on the applied inter-

polation method. Further, small-scale, convective precipitation events may not be caught by gauging networks and lead to an underestimation in precipitation.

Furthermore, hydrological models are subject to different sources of uncertainty, i.e., input, model structural and parametric uncertainty (Beven, 1993). All of the afore-mentioned uncertainties propagate to the model results and can superpose each other (Zappa et al., 2011). The overall uncertainty of hydrological models is therefore summarized as predictive uncertainty. Uncertainties are often not considered when deriving hydrological or hydro-meteorological datasets (e.g., Huang et al., 2010; Livneh et al., 2013; Zhang et al., 2014). In consequence, predictive uncertainties are often not addressed but may have substantial implications on subsequent studies, as shown by Samaniego et al. (2013). Herein, we will focus on the predictive uncertainties caused by equifinal parameter sets.

The specification of model parameters which are valid beyond catchment boundaries poses another challenge in the application of hydrologic models over large domains. Large scale hydrologic model studies apply either parameters originating from a single catchment (Henriksen et al., 2003), filter behavioral parameters from predefined sets (Perrin et al., 2008; Hartmann et al., 2015), extrapolate or regionalize parameters or hydrological variables from observed to unknown locations (Zhu and Lettenmaier, 2007; Troy et al., 2008; Xia et al., 2012b; Zhang et al., 2014) or use an uncalibrated model (Mitchell, 2004; Hostetler and Alder, 2016). A methodology considering the calibration in individual basins for creating a set of regionalized parameters which are later on filtered for behavioral solutions in all considered basins could be an alternative approach. Such an approach combines all of the afore mentioned strategies.

The aim of this study is to derive a model based, consistent set of national-scale hydrological data for Germany within the period 1951-2010. We address the need for highly resolved data by conducting observation-driven hydrological simulations at a spatial resolution of $4 \times 4$ km$^2$ (1/25°). Daily fields of evapotranspiration, soil moisture, groundwater recharge, and grid-cell-generated runoff as well as precipitation, temperatures, and potential evapotranspiration are made freely available. To our knowledge, such a consistent and long-term dataset for Germany has not been freely available until now. The dataset accounts for predictive uncertainties by considering a set of equifinal parameters. An parameter estimation approach for deriving a set of 100 parameters on the national scale is developed. We further aim to assess and evaluate the spatio-temporal distribution of the simulated hydrological states and fluxes as well as their uncertainties using multiple validation variables at different scales. Finally, the parametric uncertainties are analyzed regarding their explanatory variables for the simulated fluxes and their propagation between different model compartments.

## 2   Study Domain and Datasets

The study is conducted on the territory of Germany, which covers an area of approximately 357,000 km$^2$ (Figure 1). The region, located in Central Europe, is mainly characterized by a humid climate but nonetheless has north-to-south and east-to-west climatic gradients. The topography varies from low-altitude, flat areas in the north (North German Plain) over mid-altitude mountains in Central Germany (Central Uplands) to the high altitude Alpine Foothills and the Alps in the south. Whereas

the northwestern part of Germany is still under maritime influence, the eastern part has a more continental climate that is characterized by colder winters and less precipitation.

The assessment of water fluxes and states is restricted to the national borders of Germany because meteorological data and land surface characteristics are available in this domain. Thus, only basins entirely covered by German territory are used to derive parameters for the hydrological model. These seven major basins are depicted in Figure 1. These basins represent the topographic and hydro-climatic gradient within Germany (see Table 1). They range in size from 6,000 km$^2$ to 48,000 km$^2$ and are characterized by mean elevations ranging from 60 m.a.s.l. (Ems basin) to 560 m.a.s.l. (Danube basin). All basins have a comparable degree of urbanization ranging between 6% and 10%. A remarkably low amount of forest is observed in the Ems basin, where agriculture and pasture are the dominant land use.

Due to different climatic regimes the average streamflow of the seven basins ranges from 161 mm a$^{-1}$ to 469 mm a$^{-1}$. The low-lying Ems reaches a remarkably high discharge due to maritime influence, whereas the Saale river is characterized by the lowest streamflow. The runoff coefficient of the Saale differs significantly from the other basins, which originates from the high degree of anthropogenic influence within this basin. Three of the ten largest dams in Germany are located there (Bleiloch - 215 Mio. m$^3$, the Hohenwarte - 182 Mio. m$^3$ and the Rappbode reservoir - 109 Mio. m$^3$). Furthermore, open pit mining has a large influence on the water budget of this basin.

## 2.1 Land Surface Properties

The land surface characteristics required by the hydrologic model include a 50 m digital elevation model (DEM) acquired from the Federal Agency for Cartography and Geodesy (Federal Agency for Cartography and Geodesy (BKG), 2010), a digitized soil map at a scale of 1:1,000,000 (Federal Institute for Geosciences and Natural Resources (BGR), 1998), and a hydrogeological map at a scale of 1:200,000 (Federal Institute for Geosciences and Natural Resources (BGR), 2009). The soil map contains information on soil textural properties, such as the sand and clay contents of different soil horizons. The soils are classified into 72 soil types and have an average depth of 1.8 m. The hydrogeological map comprises 23 classes and gives information about saturated hydraulic conductivities and karstic areas. Based on the DEM, additional information, such as the slope, aspect, flow direction and flow accumulation, are inferred. Land cover information is derived from CORINE land cover scenes of the years 1990, 2000, and 2006 (European Environmental Agency (EEA), 2009). The period prior to 1990 is assumed to be static and is represented by the scene of 1990. All data sets are remapped to a common spatial resolution of 100×100 m$^2$ using a nearest neighbor approach.

The location and shape of the major basins (Figure 1) are derived via an automated delineation, which is based on gauging station and terrain information (flow accumulation and flow direction). Streamflow data are provided by the European Water Archive (EWA) (2011) and the Global Runoff Data Centre (GRDC) (2011). The results of the delineation are approved via comparison with the CCM River and Catchment Database (European Commission - Joint Research center (JRC), 2007; Vogt et al., 2007). In addition to the seven major basins (as described above), the model is set up in 222 additional, smaller basins to cross-validate the model performance.

## 2.2 Meteorological Forcings

The hydrologic model is forced with daily fields of precipitation and minimum, maximum, and average temperature. They are derived from local observations operated by the national weather service (Deutscher Wetterdienst (DWD), 2015). The station network comprises, on average, 3,800 rain gauges and 570 climate stations per year (period: 1951-2010), which have an average minimum distance of 6 km and 14 km between neighboring stations, respectively.

These local observations are interpolated on a regular grid of $4 \times 4$ km$^2$ using external drift Kriging. The terrain elevation (DEM) is used as the external drift, and the Kriging weights are based on a theoretical variogram. The variogram is estimated for all of Germany by fitting to an empirical variogram (see appendix A1). To avoid discontinuities in the interpolated meteorological forcings and consecutively in the hydrologic simulation, an estimation of multiple variograms for different climatic zones or distinct morphological regions has been rejected. The spatial resolution of $4 \times 4$ km$^2$ is seen as appropriate, considering the aforementioned station network density of precipitation observations. The quality of the interpolation is assessed by the Jacknife method (leave one out strategy) which leads to a mean relative bias of 0.64% for all precipitation stations (see appendix A2). Subsequently, daily fields of potential evapotranspiration are estimated with the Hargreaves-Samani method (Hargreaves and Samani, 1985) using interpolated temperatures (average, minimum, and maximum).

The interpolation of the precipitation is evaluated with gridded precipitation data (REGNIE) provided by the German Meteorological Service (Deutscher Wetterdienst (DWD) (2013); Rauthe et al. (2013)). The REGNIE data are based on the same observations and have a spatial resolution of 1 km. They are derived by applying a multiple linear regression approach, which accounts for daily atmospheric conditions and terrain properties, such as elevation, slope, and aspect (Rauthe et al., 2013). After remapping the REGNIE data to the aforementioned $4 \times 4$ km$^2$ grid by bilinear interpolation, a satisfactory correspondence between the interpolation and the REGNIE precipitation data is found (see Samaniego et al. (2013)). The spatially averaged bias of the daily fields is 0 with a standard deviation of 0.11 mm d$^{-1}$ within the period 1951-2010.

## 3 Methodology

### 3.1 The mesoscale Hydrologic Model mHM

mHM (www.ufz.de/mhm) is a distributed hydrologic model that accounts for the following main processes: snow accumulation and melting, evapotranspiration, canopy interception, soil water infiltration and storage, percolation, and runoff generation. These processes are conceptualized as water fluxes between internal model states similar to existing models, such as HBV (Bergrström, 1976) or VIC (Liang et al., 1994). Snow accumulation and melting processes are based on the improved degree-day method, which accounts for increased snow melting during intense rainfall events (Hundecha and Bárdossy, 2004). A three-layer discretization is used to account for the processes that represent the root-zone soil moisture dynamics. The two upper layers end in 0.05 m nad 0.25 m, and the lowest layer is spatially variable in depth depending on the soil map. On average, the lowest layer is 1.8 m deep in Germany. The evapotranspiration from soil layers is estimated as a fraction of the potential evapotranspiration depending on the soil moisture stress and the fraction of vegetation roots present in each layer.

The runoff generation in mHM is formalized as the sum of the direct runoff, slow and fast interflow, and baseflow components. The runoff generated at every grid cell is routed to the outlet using the Muskingum-Cunge algorithm. For a detailed model description, interested readers may refer to Samaniego et al. (2010) and Kumar et al. (2013b). To date the model has been successfully applied to various river basins across Europe (including Germany), the USA (Kumar et al., 2010; Samaniego et al., 2013; Kumar et al., 2013a; Thober et al., 2015; Rakovec et al., 2016; Zink et al., 2016), and worldwide (Samaniego et al., 2016).

A feature that is unique to mHM is its technique for estimating effective model parameters: Multiscale Parameter Regionalization (MPR, Samaniego et al. (2010); Kumar et al. (2013b)). Its basic concept is to estimate parameters (e.g., soil porosity) based on physiographic properties (e.g., sand and clay content) and transfer functions (e.g., pedotransfer functions). These transfer functions depend on *transfer* or *global* parameters (e.g., factors of the pedotransfer functions) that are time-invariant and location-independent. For the domain of Germany 68 *global* parameters were purpose to an automated calibration (described in section 3.2). An overview of the *global* parameters and the resulting effective model parameters can be found in the supplemental material.

This regionalization of model parameters is conducted at the high-resolution land surface property input, e.g., $100 \times 100$ m$^2$. In a second step these parameters are subsequently upscaled to the user-specified resolution of the hydrologic simulations, e.g., $4 \times 4$ km$^2$, by applying parameter-specific upscaling rules (Samaniego et al., 2010). This procedure yields in effective parameter values (e.g., soil porosity) which are used for the simulation of hydrological processes (e.g., soil water retention). Thus, the effective parameters account for the sub-grid variabilities of land surface properties, such as terrain or soil information.

## 3.2 Derivation of Representative Parameter Sets

One of the goals of this study is to derive consistent model parameters to perform nationwide simulations of water fluxes and states. A two-step parameter selection procedure was used for this purpose. In the first step, we estimate 100 sets of *global* parameters via calibration in each of the seven inner German river basins (Figure 1) independently.

In the next step, we transfer these calibrated parameter sets to the remaining basins. The parameter sets exceeding a Nash-Sutcliffe model efficiency of 0.65 (NSE $\geq$ 0.65) in all seven basins during the evaluation period (1965-1999) are retained. This parameter selection procedure ensures that the resulting ensemble parameter sets do not exhibit spatial discontinuities at basin boundaries.

The calibration is performed using the dynamically dimensioned search (DDS) algorithm (Tolson and Shoemaker (2007)). The objective function for calibration consists of an equally weighted power law function for the NSE (Nash and Sutcliffe, 1970) of the streamflow and the logarithm of the streamflow to consider high and low flows within the objective function. A compromise programming technique (Duckstein, 1984) using a power law with an exponent $p = 6$ is used to estimate the

multi-objective function ($\Phi$). This technique ensures equal improvement of the different measures $\phi_i$ during a multi-objective calibration. The overall objective function $\Phi$ is given as

$$\Phi = \left( \sum_{i=1}^{2} w_i^p \phi_i^p \right)^{\frac{1}{p}} \text{ with } \sum w_i = 1 \tag{1}$$

with

$$\phi_1 = \text{NSE}(Q) \quad = 1 - \frac{\sum_{t=1}^{T}(\widehat{Q_t} - Q_t)^2}{\sum_{t=1}^{T}(Q_t - \overline{Q})^2} \tag{2}$$

$$\phi_2 = \text{NSE}(\ln Q) = 1 - \frac{\sum_{t=1}^{T}(\ln \widehat{Q_t} - \ln Q_t)^2}{\sum_{t=1}^{T}(\ln Q_t - \overline{\ln Q})^2} \tag{3}$$

where $w_i$ is the weight ($w_1 = w_2 = 0.5$) for a particular measure $\phi_i$, $\widehat{Q_t}$ and $Q_t$ denote the modeled and observed streamflow at a time step $t$, respectively. $\overline{Q}$ is the mean of the observed streamflow over all time steps $T$.

The period of 5 years from 2000 to 2004 is chosen for model calibration. This time period reflects various hydrologic conditions ranging from a high-impact flood event in Central Europe in August 2002 to a significant drought event in 2003. The remaining 35 years of available data (1965-1999) are used for model evaluation. All simulations are conducted with a 5-year spin-up period to abrogate the influence of initial conditions.

One hundred independent calibration runs are performed for each of the seven basins (Figure 1). Using 2,000 model iterations per calibration run led to a large number of model evaluations per basin (200,000). Finally, 100 of the 700 parameters sets are retained to derive nationwide ensemble simulations of water fluxes and states at a daily resolution.

## 3.3 Validation Data

In addition to streamflow in the seven major German river basins, the model performance is evaluated against streamflow in 222 additional basins and complementary data sets including evapotranspiration, soil moisture and groundwater recharge. The cross-validation of ensemble parameter sets in basins that have not been used for parameter inference should prove the ability of the model to satisfactorily estimate streamflow in various regions of Germany with differing hydrologic characteristics.

The basins for cross-validation are distributed all over Germany and range in size from 100 km$^2$ to 8,500 km$^2$. A detailed characterization of these basins is given in Table S3 within the supplemental material. A subset of these basins contains sub-basins of seven major basins. The simulation time period is adopted for the available streamflow observations but is at least 10 years. The mean simulation time period of all 222 basins is 42 years. The streamflow estimation in these basins is evaluated using the ensemble median NSE, and its uncertainty is characterized by the range between the 5$^{\text{th}}$ and 95$^{\text{th}}$ percentiles of NSEs of the ensemble simulation.

Local evapotranspiration observations are available at seven eddy covariance towers located in Germany (Figure 1, www.europe-fluxdata.eu). Carbon and water fluxes as well as all components of the energy balance, latent heat (or evapotranspiration $E_a$), sensible heat $H$, ground heat flux $G$ and net radiation $R_n$, are measured at the towers. The energy balance is, however,

often not closed at the towers (Foken, 2008; Leuning et al., 2012) so that the observed fluxes usually underestimate the real values, which needs to be corrected before comparison with a model conserving the water balance. We apply a correction to the observed fluxes similar to Kessomkiat et al. (2013). The corrected evapotranspiration values at the eddy sites are compared with the corresponding model estimates based on the root mean squared error (RMSE), the Pearson correlation coefficient ($\rho$) and the bias.

Additionally, soil moisture observations, undertaken at eddy covariance stations, are used to evaluate modeled soil moisture. Soil moisture is measured using TDR or FDR sensors, which have a control volume of a few $cm^3$. This is much smaller than the model resolution of $100 \times 100$ m$^2$. A direct comparison between observed and simulated soil moisture may therefore be misleading due to differences in spatial representativeness and sampling depth. Here we aim to analyze the temporal dynamics of soil moisture by normalizing the respective soil moisture time series (Koster et al., 2009). The anomalies are calculated as

$$z(t) = \frac{SM(t) - \mu}{\sigma} \tag{4}$$

where $\mu$ is the mean and $\sigma$ is the standard deviation of the entire soil moisture time series $SM$ at a daily resolution. It is not possible to use deseasonalized values (normalization with monthly values) because the time periods of the available observations are too short ($\approx 6$ years). The modeled soil moisture is defined herein as the fraction of porosity, i.e., the soil water content divided by porosity.

The mHM simulation for comparing the observations at the location of the eddy covariance stations is conducted with deactivated lateral processes on a single grid cell. The model resolution ($100 \times 100$ m$^2$) is adapted to the size of the footprint of the energy flux measurements, which is typically several tens to hundreds of meters. Rather than downscaling the model results, the hydrologic processes are modeled at the resolution of the observations. The transferability of mHM across scales is presented in Samaniego et al. (2010) and Kumar et al. (2013b).

The model is evaluated with spatially distributed data, i.e., evapotranspiration and groundwater recharge, besides the evaluation of the model at the point or local scale. A remote sensing based dataset is used for evaluating the monthly modeled evapotranspiration between 2001-2010. For this purpose we used the gridded $ET$ dataset based on the Moderate Resolution Imaging Spectroradiometer (MODIS) , which was acquired from the Numerical Terradynamic Simulation Group at the University of Montana (Mu et al., 2007, 2011). The spatial resolution is approximately $5 \times 5$ km$^2$ ($0.05°$) which is close to the model resolution of $4 \times 4$ km$^2$. The evapotranspiration estimates are based on the Penman-Monteith energy balance equation using global daily temperature, actual vapor deficit, incoming solar radiation as well as remotely sensed leaf area index, fraction of photosynthetic active radiation, albedo, and land cover characteristics. The meteorological variables are based on the reanalysis product from the Global Modeling and Assimilation Office whereas vegetation products are derived from MODIS. Interested readers may refer to Mu et al. Mu et al. (2007, 2011) for detailed description of the MODIS $ET$ product.

As a second spatial dataset we utilize a long-term estimate of annual recharge over Germany (1961-1990). Due to the lack of observations, the estimated recharge from the Hydrologic Atlas of Germany (Federal Ministry for the Environment Nature Conservation Building and Nuclear Safety, 2003) is taken here as a reference. This recharge estimate is obtained

using a multiple regression model accounting for long-term estimated generated runoff, depth of the groundwater table, and regionalized baseflow indices (Neumann and Wycisk, 2003). The regionalized baseflow indices are estimated with a linar regression based on the ratio between direct runoff and total runoff as well as terrain properties, such as slope and land cover among others. Due to the various assumptions and mathematical fittings behind this recharge estimate, it is taken as an indication for model evaluation rather than an evidence. The gridded recharge estimate is available at a $1\times1$ km$^2$ spatial resolution, which is remapped to a $4\times4$ km$^2$ resolution using bilinear interpolation to be comparable to the model estimates.

## 3.4 Uncertainty of Ensemble Model Simulations

The uncertainty of the modeled evapotranspiration, groundwater recharge, grid-cell-generated runoff and soil moisture is assessed by two different criteria. First, the spatially distributed uncertainties are presented as maps showing the coefficient of variation $c_v$, which is defined as

$$c_v = \frac{\sigma}{\mu} \tag{5}$$

in which $\mu$ is the mean and $\sigma$ the standard deviation of the ensemble simulations. A large $c_v$ describes a large variation in the modeled flux or state normalized with $\mu$. $\mu$ and $\sigma$ are derived from the 100 ensemble realizations of the hydrologic model mHM on every grid cell. The variances within the ensemble simulation are caused by predictive uncertainties. These uncertainties stem from the parametric uncertainty itself and from the transfer of parameters to locations that have not been used for model calibration. In the following, the variations of the ensemble simulations are denoted as uncertainty.

Second, to assess the temporal variation of the uncertainty throughout a year, the range and normalized range of the respective flux or state are considered. The range is defined as the difference between the 5$^{\text{th}}$ ($p_5$) and 95$^{\text{th}}$ ($p_{95}$) percentiles of the ensemble simulation, whereas the normalized range is defined as

$$r = \frac{p_{95} - p_5}{p_{50}} \tag{6}$$

where $p_{50}(x)$ denotes the median value of the ensemble simulation (50$^{\text{th}}$ percentile). The 5$^{\text{th}}$ and 95$^{\text{th}}$ percentiles are chosen to exclude potential outliers from the analysis.

## 4 Results and Discussion

The model simulations are evaluated against multiple variables available at different spatial and temporal resolutions. These include daily and monthly time-series of streamflow measured at the basin outlets, soil moisture and evapotranspiration at seven eddy covariance sites, monthly fields of satellite retrieved evapotranspiration, and a long-term, annual recharge map. mHM simulations are carried out at an hourly time scale at two spatial resolutions, i.e., $100\times100$ m$^2$ at the eddy covariance stations and $4\times4$ km$^2$ at the basin level and for the nationwide ensemble simulations. Finally, an analysis of the model runs

for the nationwide water fluxes and states, including grid-cell-generated runoff ($Q_G$), evapotranspiration ($E_a$), groundwater recharge ($R$) and soil moisture ($SM$), is presented. The focus here is to provide a comprehensive overview of regional-scale water fluxes and states over Germany and analyze the uncertainty in modeled variables due to an ensemble of model parameters. The uncertainties are investigated with respect to their temporal and spatial distributions and their triggering sources. Finally, the interaction of uncertainties through the different model states and fluxes is analyzed.

## 4.1 Streamflow Evaluation in Major German River Basins

In this section we present the evaluation of mHM simulated streamflow with observations in terms of NSEs at daily and monthly timescale for a validation (1965-1999) and a calibration (2000-2004) period. Additionally, we show the hydrographs resulting from the ensemble parameter sets in comparison with observed streamflow.

The daily streamflow dynamics in the major German basins is satisfactorily captured by the model revealing a mean NSE of 0.89 and 0.84 using the on-site calibrated parameters in the calibration and validation periods, respectively (Figure 2 white boxes). The model performance is lower during the validation period in comparison to the calibration period. Such a deterioration of model performance, which is common to other hydrological model applications, is caused by differences in hydro-meteorological regimes between the calibration and validation periods (Merz and Blöschl, 2004; Merz et al., 2011) and constraining (over-fitting) of the parameters to compensate for errors in the model structure (Clark and Vrugt, 2006). Using the on-site calibrated parameter sets, the model exhibited improved performance for monthly streamflow simulations with an average median NSE of 0.97 and 0.92 during the calibration and validation period, respectively.

The ensemble parameter sets, which are depicted as the grey boxes in Figure 2, also reveal appropriate model performance. The median NSE corresponding to the ensemble parameter sets is 0.80 for daily streamflow in the validation period averaged across the seven basins. The median NSE of the ensemble parameters drops by approximately 6% compared to that of the on-site estimated parameters. This loss is reasonable considering that the ensemble parameter sets are a compromise solution, which should perform well across all seven basins (see section 3.2). The performance loss can be attributed to changes in the specific basin climatic and land-surface conditions including terrain, soil, and vegetation properties.

Changes in the predictive uncertainty corresponding to on-site and ensemble parameter sets are assessed using the range of model performance. The spread of NSEs for the monthly streamflow is considerably narrower compared to the daily flows (Figure 2). The high temporal variability of the daily streamflow is smoothed when averaged over a longer (monthly) time scale leading to an overall better correspondence between observed and simulated flows.

The ranges of NSEs corresponding to the 100 on-site and ensemble parameter sets are comparable across the investigated basins with exception of Main and Danube. In these two basins the ensemble parameter sets provided a relatively larger range of NSEs. The relatively higher spread in NSE in those basins is likely to stem from the fact that different basins are sensitive to different parameters. For example, the Ems basin, located in the maritime-influenced north, is not as sensitive to snow parameters as the alpine-influenced Danube basin. Consequently, parameters that originate from the Ems basin potentially deteriorate ensemble predictions in the Danube basins. A simultaneous calibration of multiple, distinct basins would be beneficial for deriving hydrological fluxes and states at national or continental scales.

Examples of the modeled streamflow time series are given in Figure 3. In general, the model is able to adequately capture the discharge dynamics across the investigated basins. A relatively lower mode skill in capturing the discharge dynamics in the Saale river basin can be attributed to heavy human interactions. The highly regulated streamflow in the headwaters of the Saale river (see section 2) is difficult to capture and thus leads to lower performance because mHM includes no reservoir operation. The main discharge mechanisms of Saale are considered to be adequately captured because the median NSEs are exceeding 0.85 and 0.7 at the monthly and daily resolutions for the ensemble parameter sets, respectively (Figure 2).

Interestingly, this basin shows equal or higher performance for the ensemble parameter sets compared to the on-site parameter sets in the evaluation period. A similar behavior can be observed for the Weser basin. We conclude that streamflow simulations in some basins improve by gaining knowledge from remote locations.

The Mulde basin has a tendency to underestimate peak flows (Figure 3). This could be attributed to the precipitation product. The headwaters of the Mulde basin are located in the Ore mountains at the border between Germany and the Czech Republic (Figure 1). In addition to a sparse network of rain gauges in these mountainous area, a lack of information on meteorological variables from the neighboring country (i.e., the Czech Republic) leads to an underestimation of precipitation in the interpolation process, especially for orographic-driven events. The model performance for the Mulde is comparably superior to those found by other studies, like Fleischbein et al. (2006) or Huang et al. (2010).

The results presented in this section show that the method for determining ensemble parameter sets (section 3.2) leads to satisfactory estimations of streamflow in the basins used for parameter inference. Overall, the model performance shown herein compares well to those of other studies, such as Lohmann et al. (1998); Strasser and Mauser (2001); Menzel et al. (2006); Fleischbein et al. (2006); and Huang et al. (2010). A further investigation of the applicability of the ensemble parameter sets on additional, smaller basins is shown in the following section.

## 4.2 Streamflow Evaluation at Non-calibrated Basins

Following Klemeš (1986), the model performance is evaluated across 222 basins diverging in size and geographical location. The streamflow data of these proxy locations have not been used during the model calibration. This cross-validation test focuses on evaluating the model performance against streamflow simulations along a diverse range of climatic and land-surface conditions. The evaluations shown in Figure 4 indicate a satisfactory agreement between simulations and observations. The daily streamflow simulations (Figure 4, panels A, B) reveal a median NSE value of at least 0.5 across the investigated basins based on the ensemble parameter sets. The overall average NSE value is 0.68. Expectedly, the model exhibits better skill in capturing monthly streamflow dynamics, with an ensemble median NSE averaged across all basins of approximately 0.81 (Figure 4, panels D, E). Furthermore, the ensemble median NSE exceeded a value of 0.75 in more than 20% of the basins for the daily flows and 80% for the monthly flows. The spatial variability of the median NSE across the investigated basins is low with a standard deviation of approximately 0.09 for both daily and monthly flows.

To illustrate different climatic regimes of the 222 basins, we make use of the dryness index $E_p/P$ (Budyko, 1974). Various studies describe the relationship between the dryness and evaporative index $E_a/P$ (Schreiber, 1904; Ol'dekop, 1911; Budyko, 1974; Gerrits et al., 2009) and span an uncertainty band around Budyko's curve. The model performance of the 222 basins is

plotted in panels A and D of Figure 4 using these indexes. It separates the basins into energy- ($E_p/P < 1$) and water-limited conditions ($E_p/P > 1$). The simulated evapotranspiration $E_a$ is used to derive the Budyko plot to identify potential errors in the water balance closure (Figure 4 panels A, D). All basins under investigation lie perfectly within the uncertainty ranges of the reported theoretical curves. Please note that energy limited basins are closer to the lower uncertainty line of the reported curves,

whereas water limited basins tend to the upper curve. In consequence basins with energy limitation tend to underrepresent the original Budyko curve and develop to overrepresentation for water limited locations. In conclusion, the water balances of those basins are well closed, with a mean closure error of 1% for the median simulation. The performance is comparable for basins in different climatic regimes. Such behavior is not obvious as studies like Newman et al. (2015b) and Xia et al. (2012a) found a significant dependency on the climatic regime. However, a tendency to perform better in large basins is observed. A similar

conclusion was drawn by McMillan et al. (2016).

We further analyzed the relationship between model performance and physiographic attributes (e.g, terrain or land cover characteristics). These analyses did not show any significant relationship (see Figure B1). The absence of pairwise relationships between model performance and climatic or land surface characteristics confirms the validity of the derived ensemble parameters for the national scale. In contrast, Newman et al. (2015b) and McMillan et al. (2016) observed significant depen-

15 dencies between model performance and basin characteristics, such as aridity or basin area.

The uncertainty for the individual basins caused by the ensemble parameter sets is expressed as the range between the $5^{\text{th}}$ and $95^{\text{th}}$ percentiles of NSEs (Figure 4, panels C, F). Substantial performance differences occur in 70% (45%) of the basins exceeding a range of 0.1 NSE for the daily (monthly) flow simulations. A geographical dependency of the uncertainty cannot be found as no spatial clustering is observed. Whereas daily flows show almost no relation between median NSEs and

20 the uncertainty range, i.e., worse performing basins reveal high uncertainties, the monthly NSEs show less uncertainty if the corresponding model performance is high.

The evaluation of the ensemble parameter sets presented in this section supports the hypothesis that the ensemble parameter sets are valid on the national scale. Studies like Perrin et al. (2008); Xia et al. (2012a); Cai et al. (2014); McMillan et al. (2016) and Hostetler and Alder (2016) validate their models based on streamflow over a large sample of basins and observed similar

or lower NSEs. In the following section, evapotranspiration, soil moisture, and groundwater recharge estimates are evaluated.

## 4.3 Evapotranspiration and Soil Moisture Evaluation at Eddy Covariance Stations

The ensemble model simulations are further evaluated with the evapotranspiration ($E_a$) and soil moisture ($SM$) observed at seven eddy covariance stations (Figure 1) to assess the model's ability to represent other fluxes and states next to streamflow. The ensemble median of the daily sum of evapotranspiration is plotted against the corresponding observations in Figure 5, and

30 the resulting error metrics are summarized in Table 2.

The scatter plots shown in Figure 5 indicate no systematic over- or underestimation of the observed evapotranspiration. The highest deviation in terms of RMSE is observed during summer, when the highest fluxes occur, and the lowest during winter, in which the contribution of $E_a$ is lowest among all seasons. The average bias estimated across all stations during spring is 0.34 mm d$^{-1}$, whereas it is 0.08 mm d$^{-1}$, 0.04 mm d$^{-1}$ and 0.04 mm d$^{-1}$ for winter, summer and autumn, respectively. The slight

overestimation of the modeled $E_a$ during spring is likely caused by the lack of a dynamic vegetation growth module in mHM. Thus, the onset of the vegetation period may not be captured adequately by the model. With respect to the vegetation class, the stations E1 and E6 covered by crops have the largest errors, with $E_a$ RMSEs of 19.4 mm mon$^{-1}$ and 15.4 mm mon$^{-1}$ for monthly evapotranspiration, respectively (Table 2). These errors arise because of the high impact of human interactions on croplands, e.g., due to seeding, harvesting or irrigation, compared to other vegetation classes. Additionally, the land cover class cropland is not explicitly represented within the model; rather, it is generalized within a mixed land cover class, representing all land cover types different from sealed and forest. Varying goodness of fit for different land covers and seasons for evapotranspiration at eddy flux towers were found for the four land surface models used in NLDAS (Xia et al., 2015) and thus are not uniquely observed for mHM.

In general, errors of local evapotranspiration estimates can be attributed to limitations of the Hargreaves-Samani approach for estimating the potential evapotranspiration. This approach may be inappropriate for local weather conditions. Because this method approximates the net radiation based on the minimum and maximum daily temperatures, local phenomena such as short term cloudiness, e.g., due to convective precipitation cells, are not accounted for. This effect is especially high in summer, which causes the lowest correlations between observations and simulations during this period. Unfortunately, only temperature based methods are supported by the available input data. Please notice that the observational error caused by the energy balance closure gap is, on average, 33% for the herein considered stations before applying the above-mentioned mathematical corrections.

In terms of temporal dynamics, the model is able to capture the observed evapotranspiration quite well across the different eddy covariance sites, as exemplarily shown in the upper panel of Figure 6. The model is able to adequately represent the observed monthly dynamics with an average correlation of approximately 0.93 (Table 2). The correlation between the observed and the simulated daily evapotranspirations is at least 0.77, with the exception of the cropland site E1.

The lower panel of Figure 6 shows the performance of mHM in representing the daily soil moisture anomalies, which are generally in good correspondence with observations. The temporal dynamics of observed soil moisture anomalies during the wetting and drying phases are well captured by the model. The resulting correlation shown in Table 2 at different eddy stations ranges between 0.53 and 0.93. These correlations were similar to those of other studies, such as Cai et al. (2014). The lowest values are observed at cropland sites, which is due to the above-mentioned human interaction and land cover class representativeness. The amplitude of the observed soil moisture anomalies is adequately captured by the model. Still, some peaks are not reproduced satisfactorily, which could be due to the non-representativeness of the $100 \times 100$ m$^2$ model grid cell for TDR/FDR soil moisture measurements. Thus, the simulated soil moisture is smoother compared to the observation because it represents the effective soil moisture of the entire grid cell.

## 4.4 Evaluation with Spatially Distributed Data

In this section, we present results of the model skill in representing gridded fluxes over the entire German domain. The first comparison is conducted for the assessment of reproducing the monthly fields of modeled $ET$ against the remotely sensed MODIS $ET$ product. The results are summarized in Figure 7 in terms of three key metrics: relative bias, correlation and root

mean square error (RMSE). The analysis is conducted using the ensemble mean of $ET$ from the 100 model simulations. The modelled $ET$ is able to adequately capture the spatio-temporal features of the MODIS derived ET product with the majority of grid cells (74%) having a relative absolute bias of less than 10%. Notable differences among these two $ET$ datasets are appearing in lowland areas along the Danube river basin in South Germany, where the modeled $ET$ exhibited a dry bias

5 compared to the MODIS $ET$. An opposite trend of postive bias in modeled $ET$ is observed for grid cells lying along the coastal region in north Germany. The temporal correspondence between both evapotranspiration datasets is also remarkably high with an average Pearson correlation coefficient of 0.96 (standard deviation 0.02). Notably both $ET$ datasets exhibit pronounced seasonal variability leading to a high temporal correspondence between them.

 The second assessment evaluates the modeled groundwater recharge with long-term annual values from the Hydrologic Atlas

10 of Germany (HAD) (Federal Ministry for the Environment Nature Conservation Building and Nuclear Safety, 2003). mHM's long-term recharge estimate implicitly represents the baseflow component of the total runoff based on the assumption that the underground basin is closed and that there are no external losses (e.g., irrigation or pumping). Consequently, this analysis serves as a proxy for assessing the model skill for partitioning the total runoff into interflow and baseflow. The comparison of the spatial pattern of the recharge shows good accordance between the two maps with a correlation coefficient of approximately

15 0.8 (Figure 8). The spatial pattern of the recharge follows the known climatology of Germany with high recharge rates being observed in areas with high precipitation amounts (e.g., Alps - region 11 in Figure 10).

 There are some significant differences between the modeled and HAD groundwater recharge, particularly at cells character-ized by urbanization (i.e., Munich, Hamburg, Berlin, and the metropolises of Ruhrgebiet in the northwest). The model tends to underestimate the HAD recharges, with differences as high as approximately 200 mm a$^{-1}$. Notably, the herein used version

20 of mHM treats sealed areas as almost impermeable, which is unrealistic. This issue has been resolved in recent mHM versions (5.0 and higher). In general, the HAD estimate of recharge is, on average, 31 mm a$^{-1}$ higher compared to the ensemble mean simulation. This mismatch arises from the differences in potential evapotranspiration ($E_p$), which were used for both estimates. The $E_p$ estimates used for the HAD (Federal Ministry for the Environment Nature Conservation Building and Nuclear Safety, 2003) are lower than those used for mHM simulations and result in higher water amounts remaining in the underground.

25 Besides these mismatches, the spatial pattern of the modeled groundwater recharge compares well with the HAD estimates (Figure 8).

## 4.5 Spatial Patterns of Ensemble Means and Uncertainties

The estimated evapotranspiration ($E_a$) and grid-cell-generated runoff ($Q_G$), as well as their uncertainty, which is expressed as the coefficient of variation of the ensemble simulations, are presented in Figure 9. In addition to these simulation results,

30 Figure 9 shows the mean annual precipitation, dryness index and land surface properties, i.e., porosity and dominating land cover type. Thus, Figure 9 is used to analyze the spatial patterns of uncertainty and their main causes.

 The high precipitation amounts above 1000 mm a$^{-1}$ in panel A correspond to mountainous areas in Germany. The driest region is located in the northeastern part of Germany. This is, on the one hand, due to its distance to the sea (continental climate) and, on the other hand, due to the Central Uplands in the western and central part of Germany. These mountains, especially

the Harz mountains (center of Germany), capture most of the precipitation events brought from the west. The low amounts of precipitation in the east lead to lower amounts of evapotranspiration (Figure 9, panel B) and grid-cell-generated runoff (Figure 9, panel C) in this region compared to the rest of Germany. Thus, the northeastern part of Germany is characterized by high dryness indexes of 1.2 and above. The uppermost dryness indexes up to 1.4 are located in the lee of the Harz mountains.

The average dryness index in Germany is 0.98. Another region characterized by high dryness indexes is the Upper Rhine Valley, which is known to have a locally warmer climate compared to its neighboring regions. Mountainous regions are characterized by stronger energy limitation due to high precipitation amounts, which results in dryness indexes lower than 0.65.

   The spatial distribution of the uncertainty, i.e., the coefficients of variation (see section 3.4), of the grid-cell-generated runoff (Figure 9, panel G) is mainly governed by the dryness index (Figure 9, panel D). The Spearman rank correlation between
both variables is 0.92. The uncertainty patterns of evapotranspiration (Figure 9, panel F) have a closer relation to soil textural properties, i.e, porosity (Figure 9, panel E), with a Spearman rank coefficient of 0.58 as compared to the dryness index (rank correlation=0.28). Locations of high uncertainty in $E_a$, e.g., northern Germany, correspond to regions of high porosities. Within this region, soils are dominated by sand and are highly conductive, which results in low water holding capacities. The modeled evapotranspiration is highly dependent on the soil parameterization because soil water is the main source of evaporative water.
In contrast, the uncertainty patterns of grid-cell-generated runoff, e.g., ($Q_G$) in the northeastern part of Germany and the Upper Rhine Valley, correspond to high values in the dryness index in those regions.

   In conclusion, the spatial distribution of the uncertainty in evapotranspiration is influenced by the parameterization of the soil whereas the runoff uncertainty pattern is dominated by the dryness index. The patterns appearing in the evapotranspiration and grid-cell-generated runoff at the location of big cities (orange areas in panel H of Figure 9) are caused by the above-mentioned
old representation of sealed areas for mHM versions prior to 5.0.

### 4.6  Spatio-temporal Distribution of Uncertainties

This section focuses on the spatio-temporal differences of uncertainties caused by the 100 ensemble parameter sets. Figure 10 shows the climatological dynamics and the normalized ranges (see section 3.4) of the respective variables, i.e., evapotranspiration ($E_a$), soil moisture ($SM$), groundwater recharge ($R$) and grid-cell-generated runoff ($Q_G$). The rows refer to different
environmental zones in Germany (Federal Environmental Agency, 2005), which are depicted in the upper right corner of Figure 10. For comprehensibility only a selection of 5 environmental zones is depicted therein, representing the region of high dryness indexes in the north (zone 2), Central Germany including Central Uplands (zones 4 and 9), the foothills of the Alps (zone 10) and the Alps (zones 11).

   The magnitude of the evapotranspiration uncertainty, i.e., the uncertainty range, is lowest among the four variables. Evapo-
transpiration is estimated by scaling the potential evapotranspiration with the water availability in several reservoirs, i.e., the interception storage, the surface ponds in sealed areas and the soil moisture. Notably, most of the area in Germany are characterized by humid and continental climate where the $ET$ is constrained by available energy. The evapotranspiration is thus mainly driven by the potential evapotranspiration. A relatively large uncertainty in soil moisture does not directly propagate to evapotranspiration uncertainty. The highest uncertainties are observed for the groundwater recharge. This model's internal

variable is neither closely related to the model input as $E_a$ nor indirectly constrained by calibration as the generated streamflow. In consequence, its uncertainty is highest among the four variables.

The evapotranspiration uncertainty shows almost no dynamics during the course of the year. In contrast, the uncertainty of recharge and runoff streamflow change significantly during the course of the year. Whereas the dynamics of the groundwater recharge and its uncertainty are positively correlated, the correlation for soil moisture and its uncertainty is negative. Thus, the recharge uncertainty is lowest for low recharge values, which occur in summer when the subsurface reservoirs are comparably dry. The low amplitude of the soil moisture uncertainty is reasoned in the high persistence of soil moisture. Regions of high porosity and low dryness indexes in northern Germany have more distinct dynamics compared to southern locations. The uncertainty of the generated runoff is a composite of the dynamics of soil moisture and recharge and thus shows the distribution of water among the model's internal reservoirs.

## 5 Summary and Conclusion

In this study, we present the derivation and evaluation of a high-resolution ($4\times4$ km$^2$) dataset of hydrologic and meteorological fluxes and states for Germany covering the period 1951-2010, which is freely available. The dataset incorporates 100 spatially consistent ensemble simulations, which are analyzed regarding their uncertainty caused by the parameter estimation. The parameter sets of the ensemble simulations are determined by a two-step parameter selection method. The model is calibrated in seven basins, and the parameter sets are filtered based on the cross-validation results in all of the basins. Thus, the uncertainty is composed of the uncertainty in parameter estimation and the uncertainty stemming from transferring these parameters to remote locations. The ensemble simulations are evaluated with streamflow, evapotranspiration and soil moisture observations and recharge data.

A comparable study by Newman et al. (2015a) focuses on the provision of a 100 member ensemble dataset which is focusing on meteorological variables for major parts of North America. Similar to the study presented herein they evaluate the data in a large sample of basins, i.e., 671. We, however, conclude that 100 realizations is an appropriate sample size for an uncertainty assessment study.

The evaluation regarding streamflow at 222 additional basins revealed a median NSE of 0.68. Thus, the 100 ensemble parameter sets are considered to be representative for Germany. The evaluation with evapotranspiration from eddy covariance stations showed deficiencies in mHM. Especially in spring, deviations of the modeled and observed $E_a$ indicate room for improving the representation of vegetation dynamics within mHM. The sites covered by cropland showed the largest deviations from evapotranspiration observations because croplands are highly human-influenced (seeding, harvest, or eventually irrigation), which makes it difficult to model their dynamics at the local scale. Additionally, cropland is generalized in a mixed land cover class in mHM. Soil moisture estimations at the same locations have been in good agreement with the observed dynamics.

The second part of the study focuses on the uncertainty of the simulated hydrological fluxes and states due to uncertainties in parameter estimation. It is shown that uncertainty varies in time, location and magnitude between hydrological variables. Among all of the variables, the uncertainty was lowest for evapotranspiration and highest for recharge. The spatial distribution

of runoff uncertainty is closely related to the spatial distribution of the dryness index. In contrast, the uncertainty patterns of evapotranspiration estimates are mostly connected to soil properties. In general, the highest uncertainties occur in the north-eastern part of Germany, which is characterized by low precipitation amounts and high soil porosities. The temporal variation of uncertainties is almost constant for evapotranspiration, medium for grid-cell-generated runoff and soil moisture and high for

groundwater recharge and depends on geographical location.

Based on these results we suggest incorporating additional data, e.g., in-situ soil moisture or satellite observations, into the calibration procedure to better constrain the model's internal states. The results of this study emphasize the importance of the considering parametric uncertainty for historical analysis, now- and forecasting in hydrology.

## 6   Data Availability and Data Format

The dataset consists of daily values of precipitation and minimum, maximum and average temperature, potential evapotranspiration, evapotranspiration, soil moisture, groundwater recharge and generated runoff. Whereas the latter four are provided as ensemble of 100 simulations. The data format is the Net Common Data Format (NetCDF version 3) and is based on the CF conventions (www.cfconventions.org). Additionally, the ensemble means and standard deviations are provided for download. The dataset is freely accessible under Creative Commons license at http://www.ufz.de/index.php?en=41160.

**Appendix A: Interpolation of Meteorological Data**

### A1   Variogram Estimation

The variogram for the German domain is estimated based on two different approaches. In the first approach regionalized variograms for rectangular sub-domains (blocks) were estimated (Figure A1). The interpolation of meteorological variables based on these regionalized variograms, however, lead to discontinuous fields of these meteorological variables. This result contra-

dicted the aim of deriving seamless fields of hydro-meteorological fluxes and states for entire Germany. In consequence, continuous meteorological interpolations have been the prerequisite for the next approach. In the second approach, a compromise variogram for entire Germany is estimated by considering all available data from all meteorological stations, e.g., approximately 5700 stations for precipitation, for the estimation of an empirical variogram. An exponential, theoretical variogram is fitted to this empirical variogram. The fitted variogram curves of both methods are presented exemplarily for precipitation

in Figure A1. The empirical variogram is well represented by the theoretical variogram with a root mean squared error of 0.02. The consecutive estimation of meteorological fields is based on the second approach using a compromise variogram for Germany.

## A2   Interpolation Error

The interpolation error was assessed by a leave-one out strategy, i.e., the Jacknife method. This cross-validation informs about the ability of the external drift kriging to estimate meteorological variables at locations where observations are available. The algorithm is as follows:

1. Exclude one station from the set of observations.

2. Estimate the meteorological time series at this location using external drift kriging.

3. Compare the interpolated time series with the observation and assess the interpolation error at each station.

4. Interpolate the Jacknife-error estimates over the Germany domain using ordinary kriging to obtain error maps for visualization purposes.

10   The error at each station is characterized by the bias, relative bias, root mean squared error, and Pearson correlation coefficient (Figure A2). Exemplarily we present the errors of the precipitation interpolation because this variable has the highest spatial and temporal variability among the interpolated variables (precipitation; minimum, maximum, and average temperature). The average and the standard deviation for the different errors assessments over all stations are 0.01 and 0.15 mm d$^{-1}$ for the bias, 0.64% and 5.60% for the relative bias, 0.93 and 0.03 for the Pearson correlation coefficient, and 1.75 and 0.48 mm d$^{-1}$ for the 15   root men squared error. Reviewing these values the chosen interpolation approach is seen appropriate.

## Appendix B:   Relation of Model Performance and Land Surface and Hydro-climatic Characteristics

The analysis for identifying relations between land surface and hydro-climatic characteristics and model performance is presented in Figure B1. This analysis does not reveal any hydro-meteorological or morphological conditions which explain different model performance in distinct basins. In conclusion, the retrieved parameter sets are representative for various climatic 20   and physiographic conditions.

*Acknowledgements.* We kindly acknowledge our data providers the German Meteorological Service (DWD), the Joint Research Center of the European Commission, the European Environmental Agency, the Federal Institute for Geosciences and Natural Resources (BGR), the Federal Agency for Cartography and Geodesesy (BKG), the European Water Archive, and the Global Runoff data Centre. Further, we acknowledge the projects EuroFlux (EU-FP4), CarboEuroFlux (EU-FP5), and CarboEuropeIP (EU-FP6), IMECC (EU-FP6) as well as 25   Christian Bernhofer, Axel Don, Mathias Herbst, Alexander Knohl, Olaf Kolle, and Corinna Rebmann for the provision of eddy covariance data. This work was funded by the Helmholtz Alliance - Remote Sensing and Earth System Dynamics (HGF-EDA) and the Water and Earth System Sciences Competence Cluster (WESS). It is part of the Helmholtz Alliance Climate Initiative (REKLIM) and was supported by the Helmholtz Interdisciplinary Graduate School for Environmental Research (HIGRADE). We thank three anonymous referees and the Editor, Erwin Zehe, for their comments, which helped us to improve the quality of the manuscript.

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

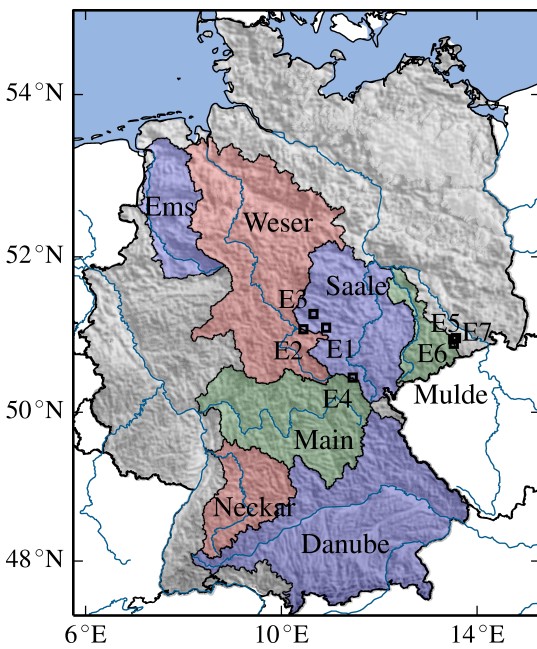

**Figure 1.** Study area showing the seven basins used for estimation of the ensemble parameter sets for Germany. The different colors are making the basins better distinguishable. The points E1-E7 denote eddy covariance stations which are used for the evaluation of evapotranspiration and soil moisture.

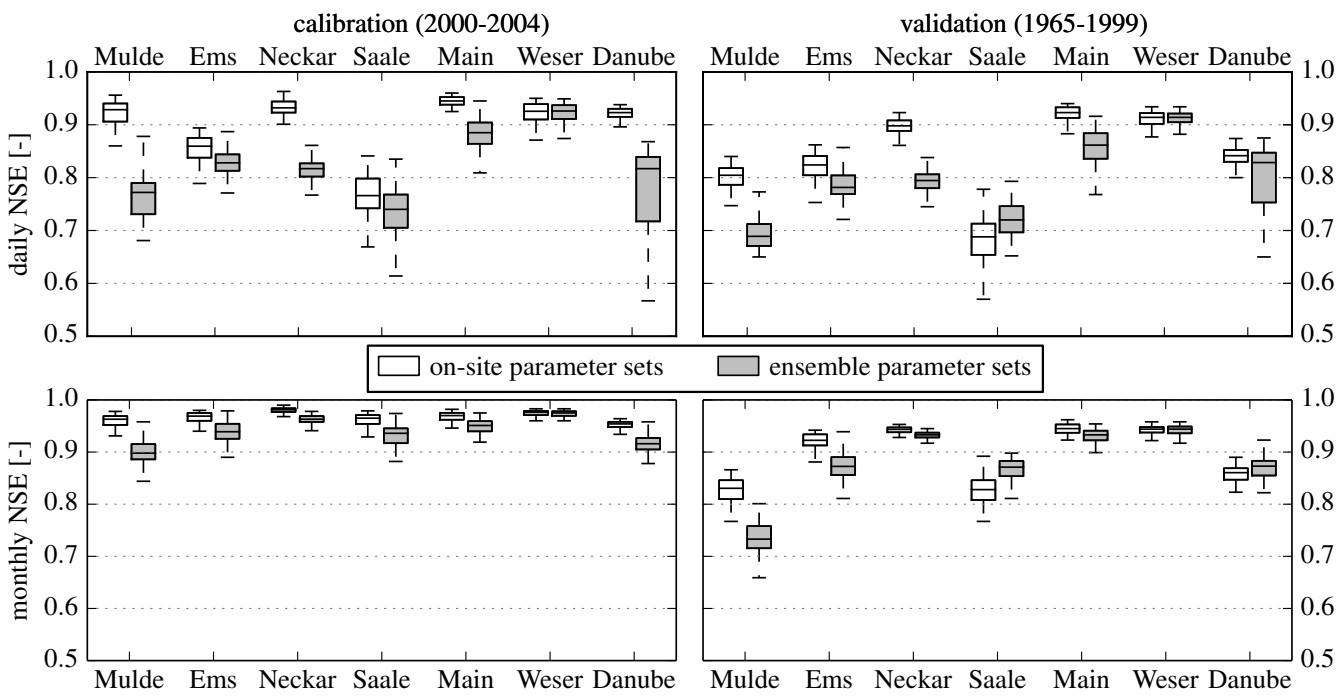

**Figure 2.** Model performance expressed as Nash Sutcliffe Efficiency (NSE) at daily (upper row) and monthly (lower row) resolutions for the calibration period 2000-2004 (left-hand side) and validation period 1965-1999 (right-hand side). The white box plots show the results of the on-site calibration, whereas the gray box plots are simulations using the 100 ensemble parameter sets for Germany. Please note that the y-axis starts at NSE=0.5

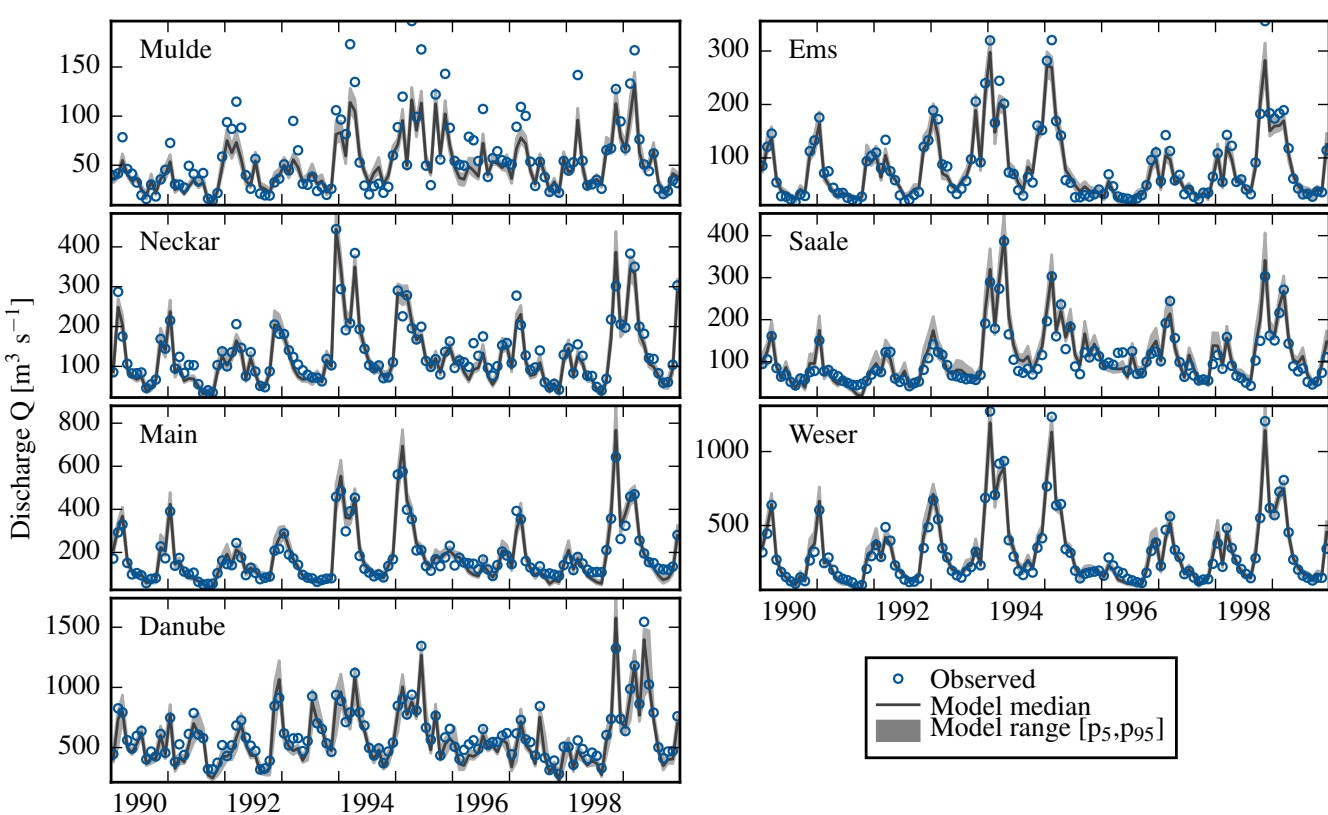

**Figure 3.** Observed and modeled monthly streamflow for the seven basins, which were used for parameter inference. The figure shows one decade (1990-1999) of the evaluation period. The solid dark gray line depicts the median model results and the light gray band depicts the range between the 5th and 95th percentile of the 100 ensemble simulations.

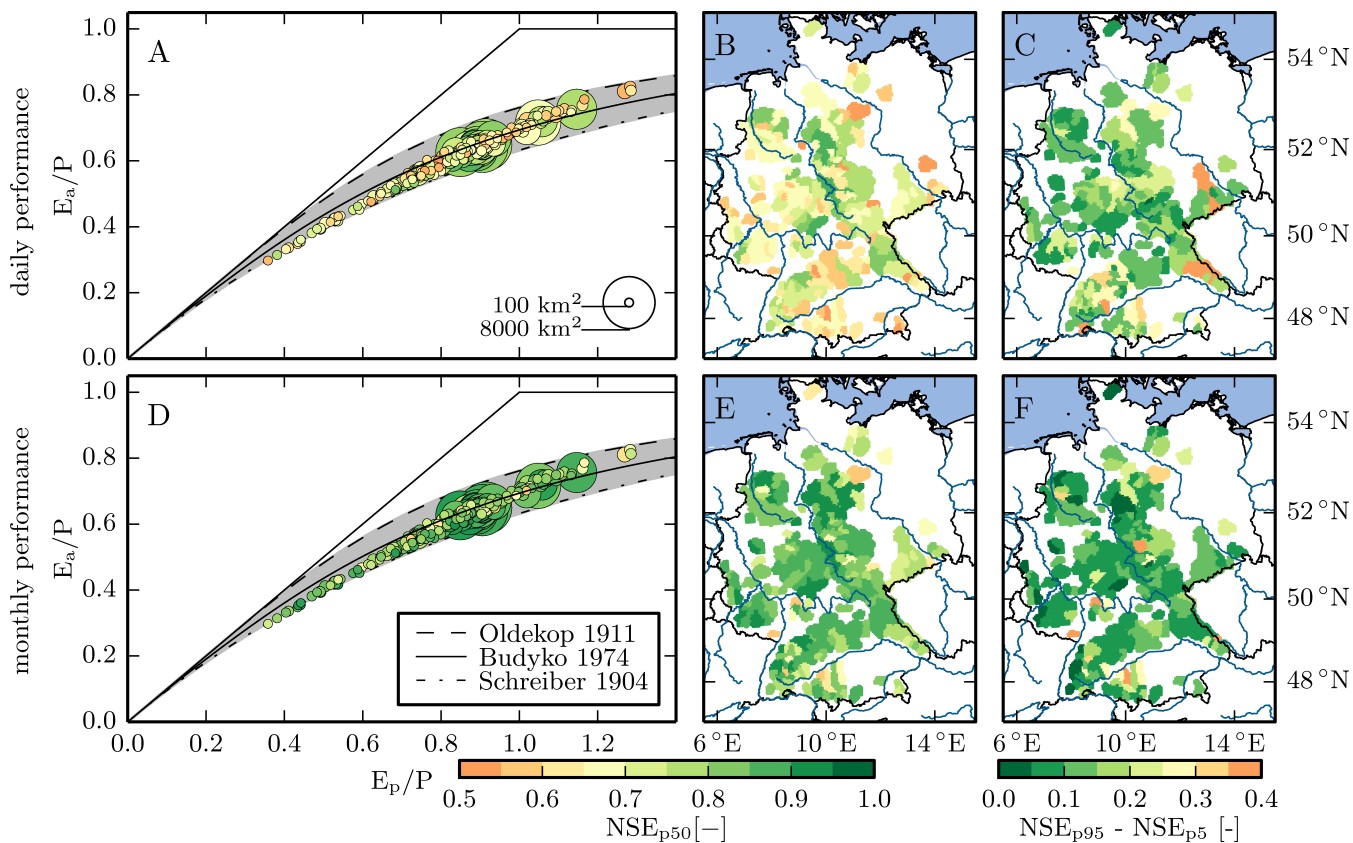

**Figure 4.** Budyko plot and performance maps for 100 ensemble parameter sets at 222 basins spread over Germany. The upper row depicts evaluations based on daily values (panels A, B, and C), whereas the lower row depicts monthly streamflow evaluations (panels D, E, and F). In the first column the basins are presented as Budyko plots (panels A and D), which are color-coded based on the ensemble median NSE for daily (panel A) and monthly (panel D) streamflow values. The gray band envelops different estimations of the Budyko curve (Schreiber, 1904; Ol'dekop, 1911; Budyko, 1974). A separation to energy- ($E_p/P < 1$) and water-limited basins ($E_p/P > 1$) can be made based on the x-axis. The center column depicts the location of the 222 basins shown in the Bydyko plots using the same color code (panels B and E). The right column shows the range of the 5[th] and 95[th] ensemble percentiles for the NSE on daily (panel C) and monthly (panel F) basis. Panels A, B, D, and E share the left color bar, and panels C and F share the right color bar. The simulation period is adopted according to the available streamflow observations but is at least 10 years (average=42 years).

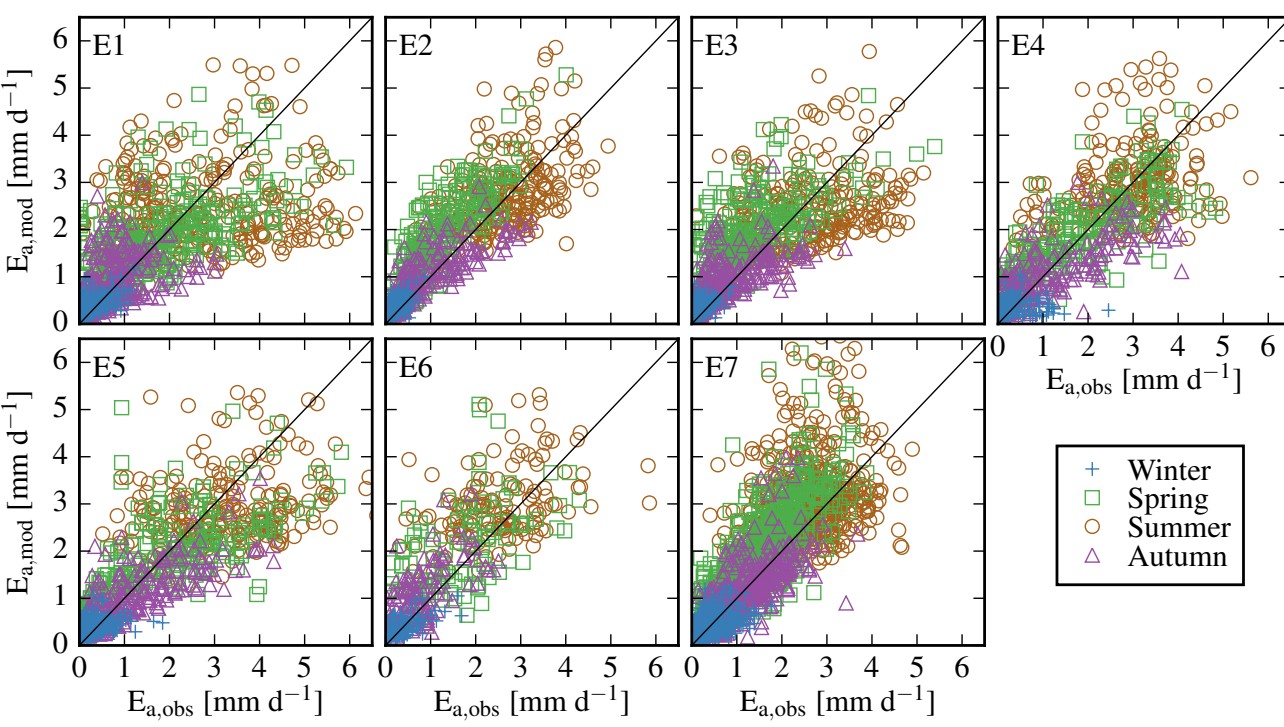

**Figure 5.** Observed ($E_{a,obs}$) versus ensemble median modeled evapotranspiration ($E_{a,mod}$) on daily basis at the seven eddy covariance stations (Figure 1, Table 2).

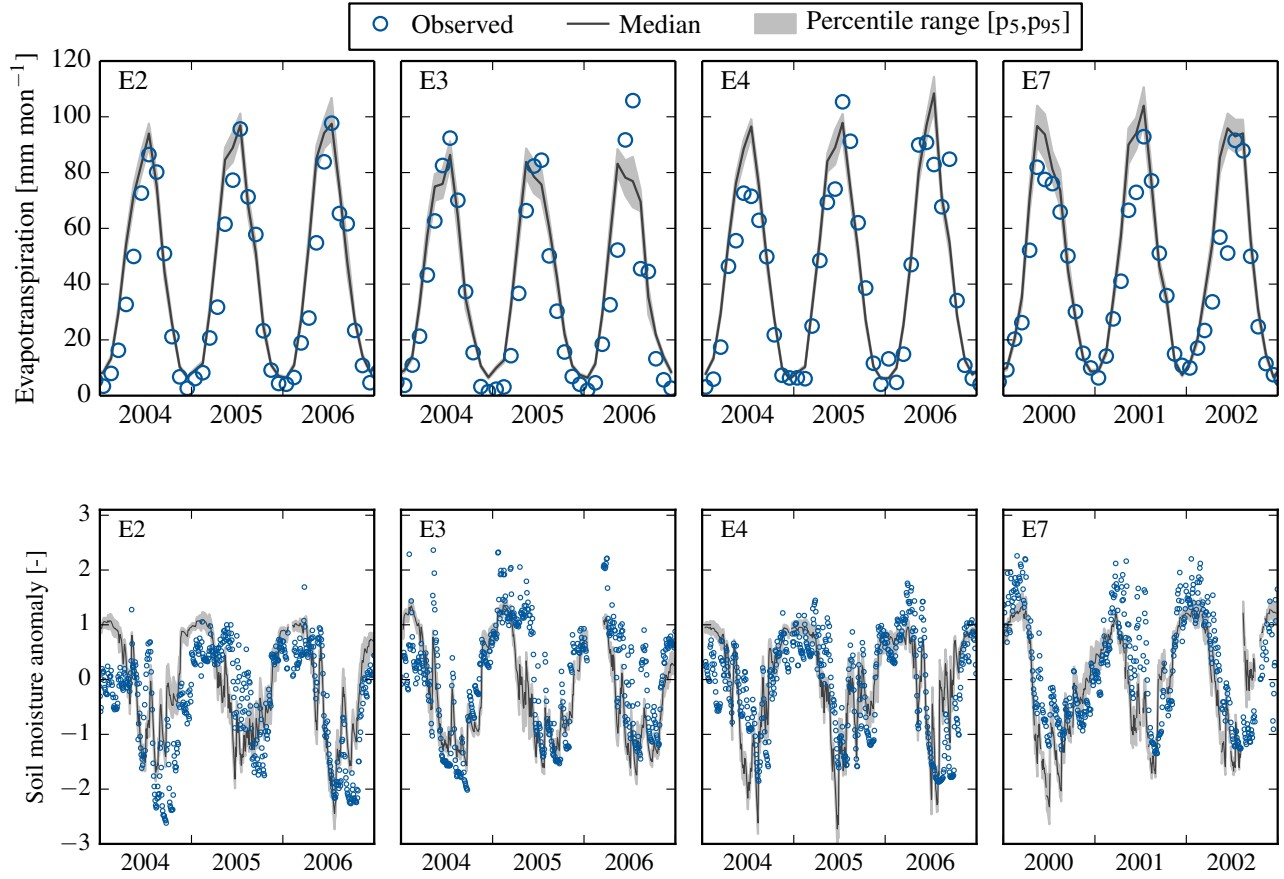

**Figure 6.** Exemplary time series of observed and modeled monthly evapotranspiration and daily soil moisture anomalies at four eddy covariance stations (Figure 1, Table 2). The four stations are chosen because they represent the two major mHM land cover classes (forest and mixed) and have three consecutive years of data without significant data gaps. Further the four station are spread over the three regions where eddy covariance observations are available. The solid dark gray line depicts the median model results and the light gray band depicts the range between the $5^{th}$ and $95^{th}$ percentile of the 100 ensemble simulations.

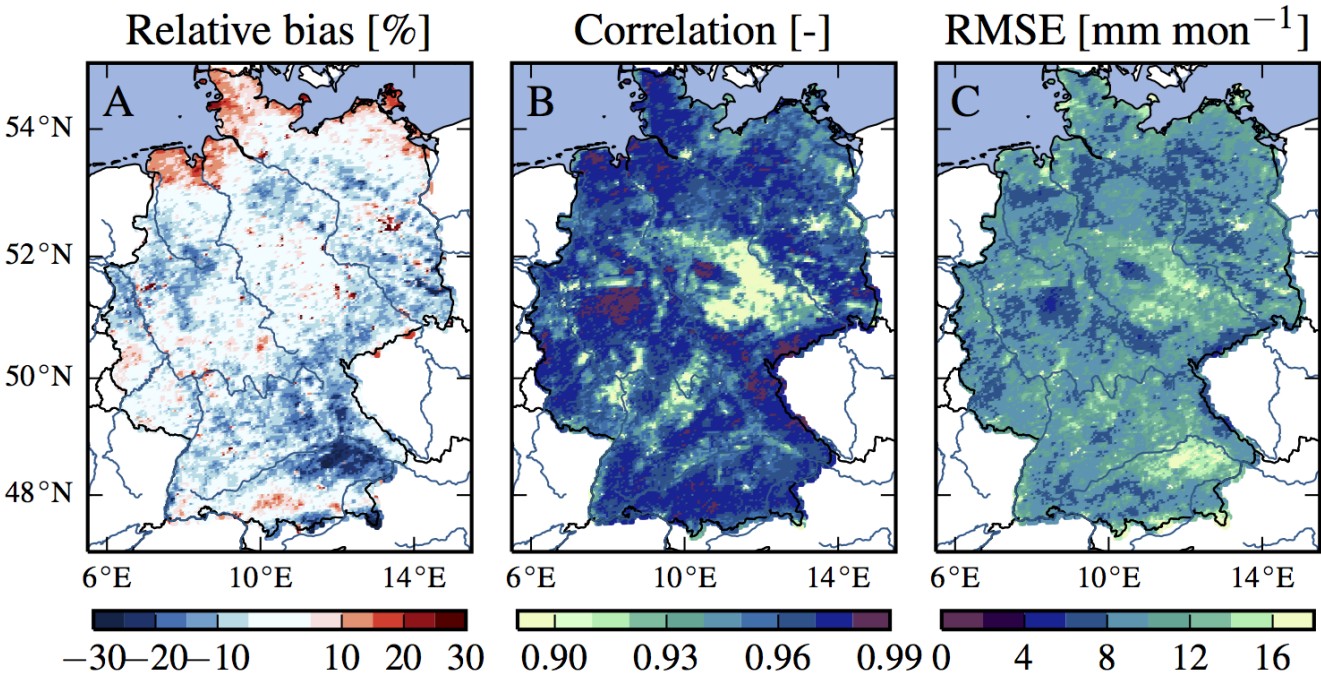

**Figure 7.** Comparison of monthly estimates of evapotranspiration from mHM and MODIS in the period 2001-2010. The ensemble is represented by the ensemble mean of 100 evapotranspiration estimates. The comparison is based on three metrics: A) relative bias, B) Pearson correlation coefficient, and C) root mean squared error (RMSE). The respective units are given in brackets.

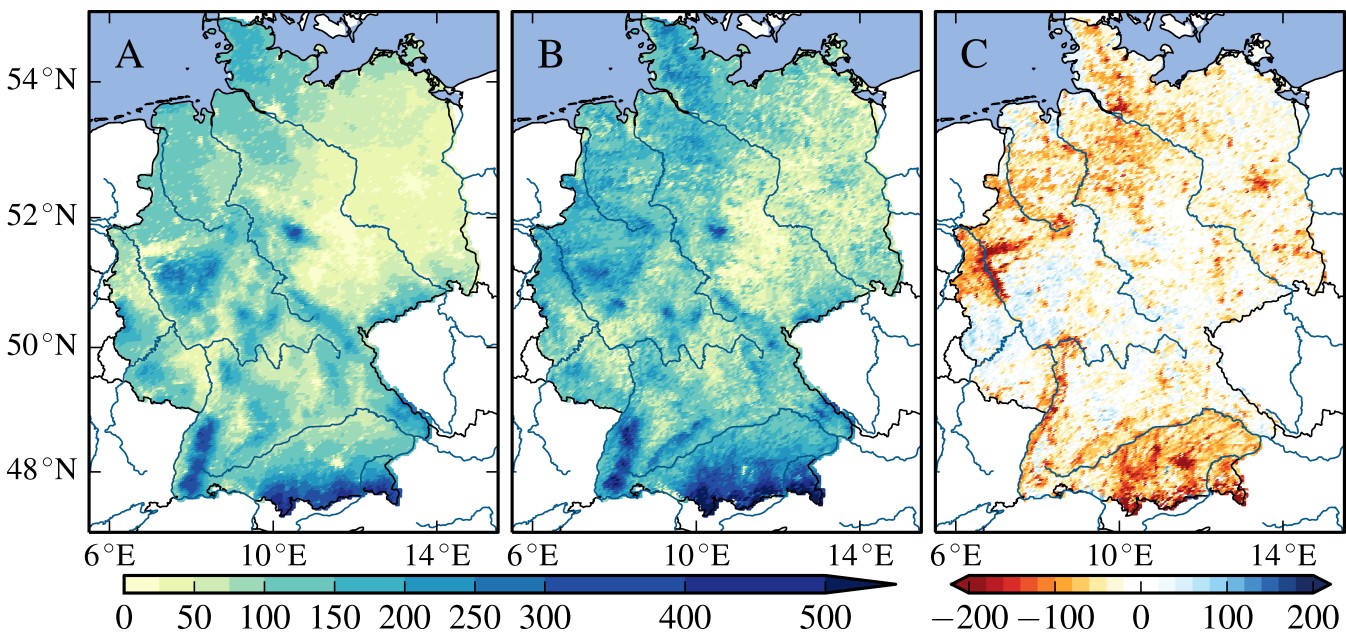

**Figure 8.** Comparison of mean annual groundwater recharge ($R$) modeled with A) mHM and from B) the Hydrologic Atlas of Germany (Federal Ministry for the Environment Nature Conservation Building and Nuclear Safety, 2003; Neumann and Wycisk, 2003). Panel C shows the difference (A-B) between the two data sets. The units are [mm a$^{-1}$] for all panels.

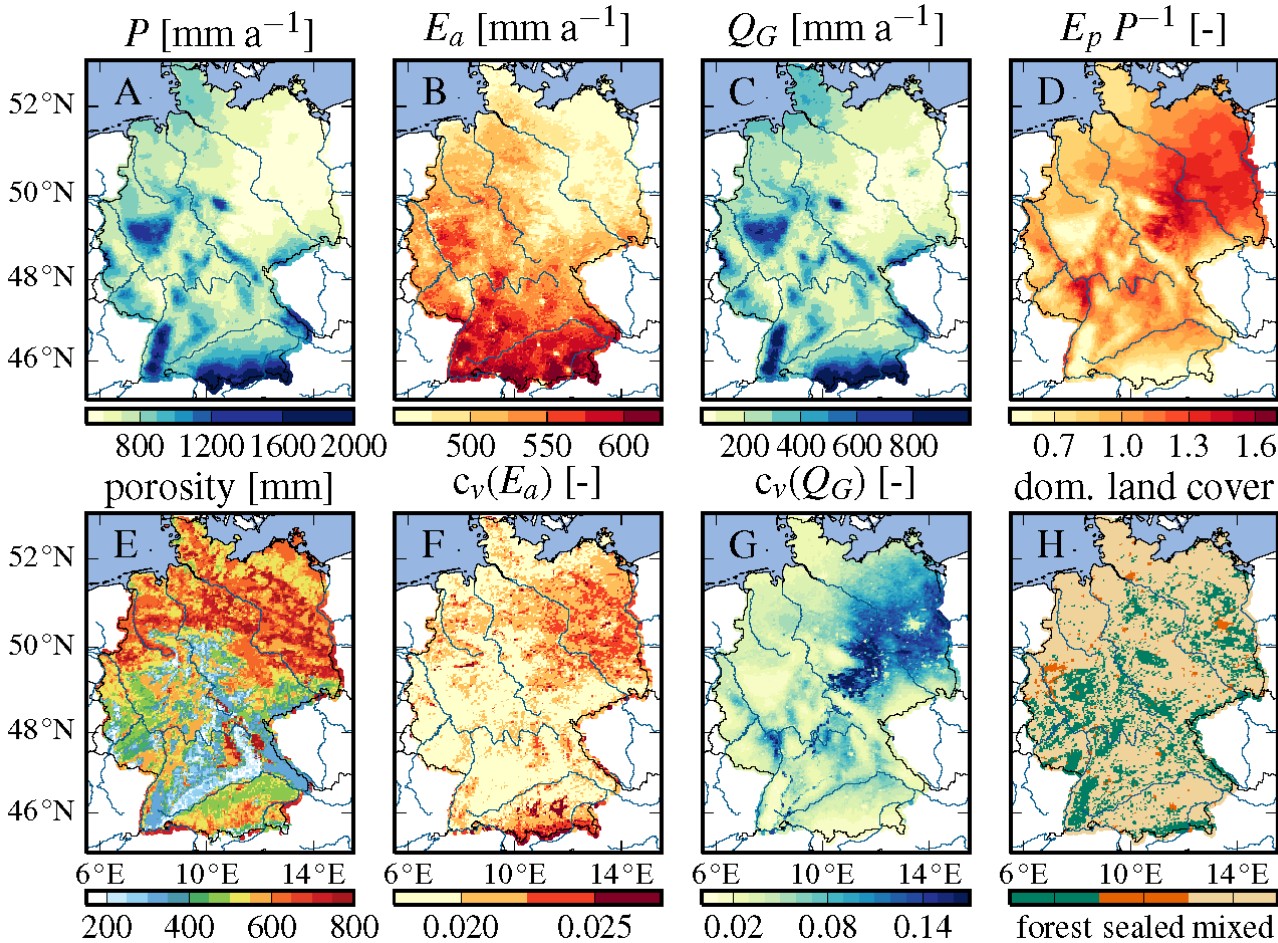

**Figure 9.** Water balance variables, their coefficients of variation and land surface characteristics for Germany. A) Mean annual precipitation $P$, B) ensemble mean annual evapotranspiration $E_a$, C) and grid-cell-generated runoff $Q_G$, D) dryness index $E_p/P$, E) sum of porosities (saturated soil water content) of all model layers, F) coefficient of variation of the ensemble of annual evapotranspiration and G) generated runoff, H) dominating land cover class on a $4{\times}4$ km$^2$ grid. The mean values and coefficients of variation are based on the period 1951-2010.

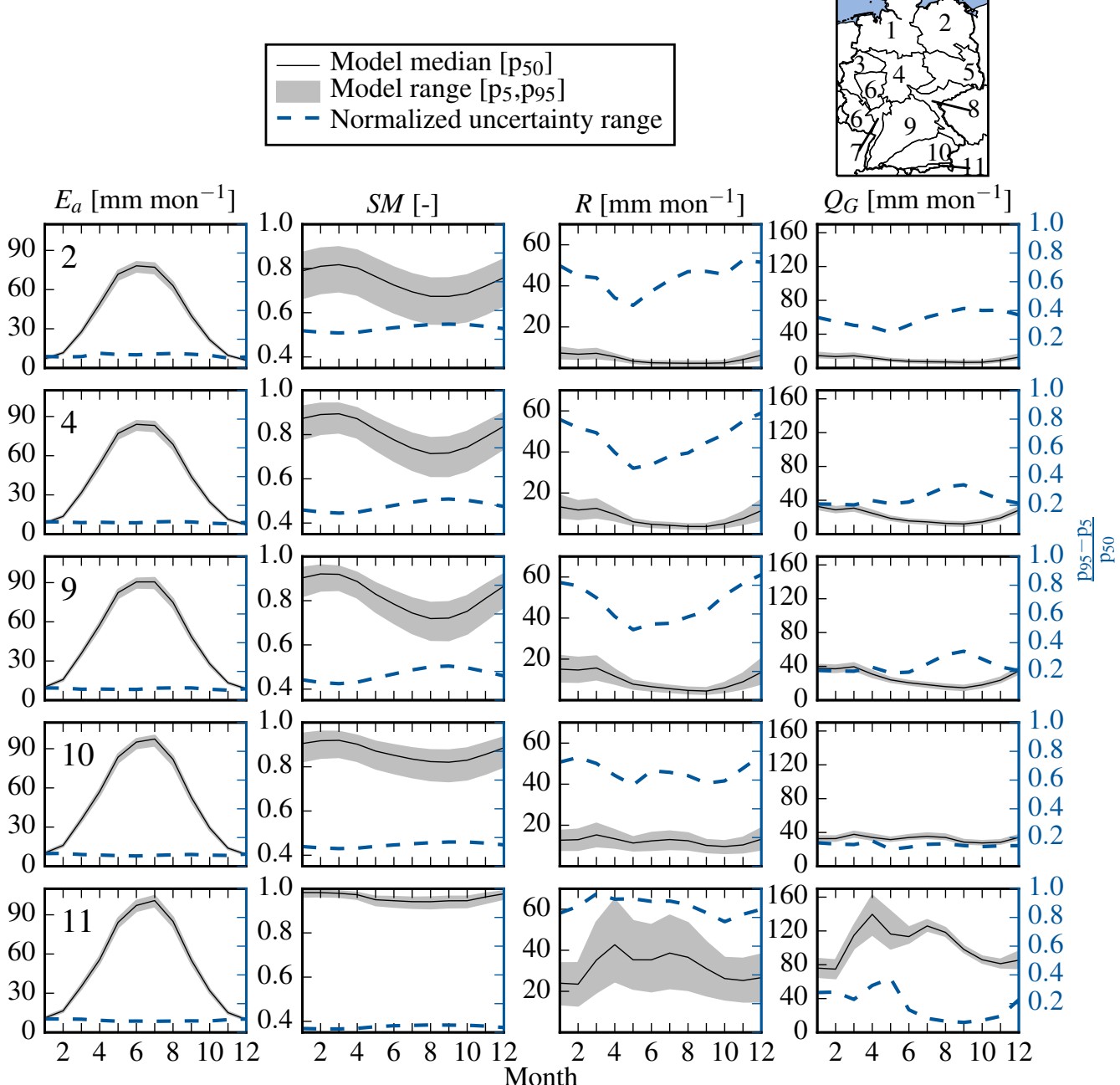

**Figure 10.** Spatio-temporal patterns of uncertainty for five different environmental zones in Germany. The locations of the different zones are depicted on the map on the upper right. The presented hydrologic variables are evapotranspiration ($E_a$), soil moisture ($SM$), recharge ($R$), and grid-cell-generated runoff ($Q_G$). The uncertainty ranges and the ensemble median refer to the left ordinate (black and grey), whereas the normalized uncertainty range refers to the right ordinate (blue). The reference period for the climatological values is 1951-2010.

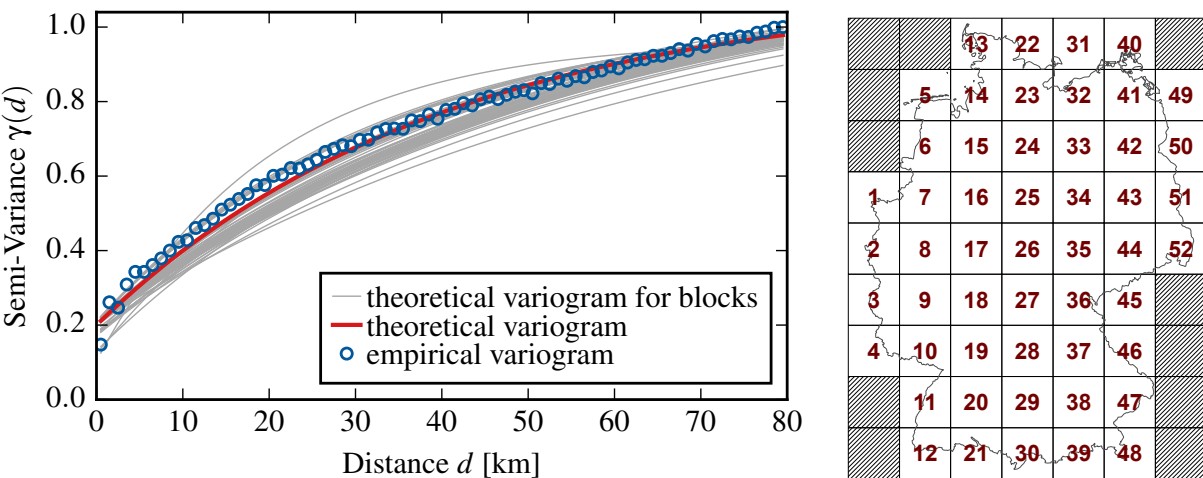

**Figure A1.** The panel on the left shows the empirical variogram (blue circles) and a fitted exponential variogram (red curve) for the entire domain of Germany as well as fittings for sub-domain (block) variaograms (grey lines). The 52 sub-domains (blocks) are depicted in the panel on the right.

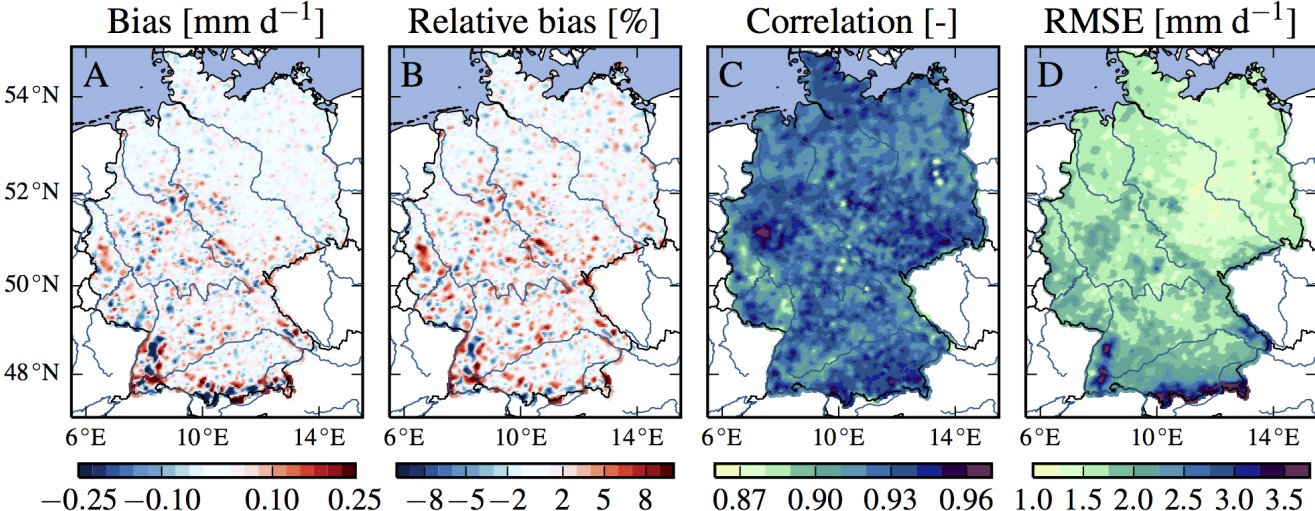

**Figure A2.** Evaluation of the interpolation at precipitation stations based on a leave one out cross-validation strategy, i.e., the Jacknife method. The performance criteria from the individual stations are interpolated to a $4\times4$ km$^2$ grid using ordinary kriging. The panels denote different performance metrics: A) bias, B) relative bias, C) Pearson correlation coefficient, and D) root mean squared error (RMSE).

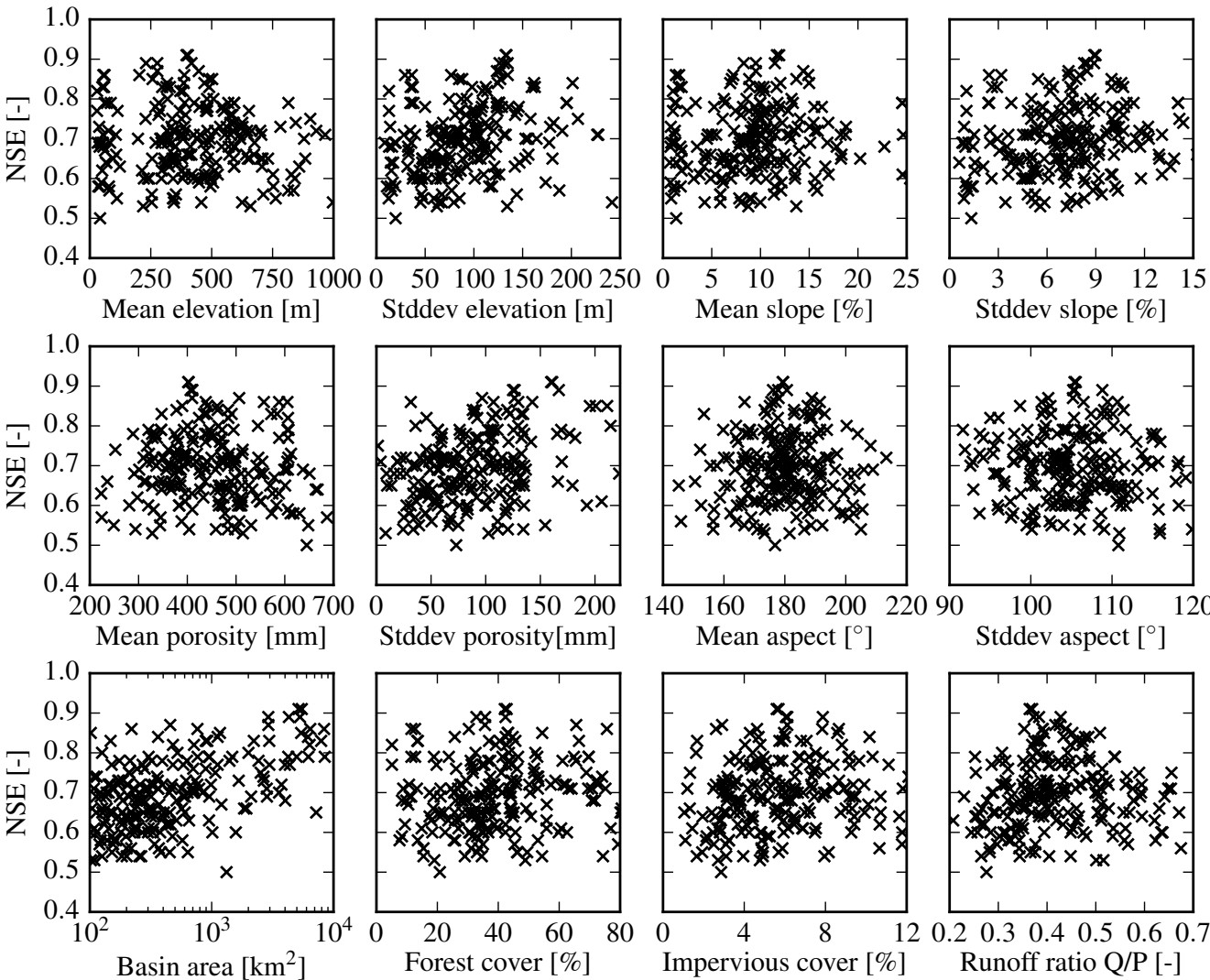

**Figure B1.** Relation between land surface and hydro-climatic conditions and model performance for the 222 river basins. The location of the basins is depicted in Figure 4. The mean and standard deviation (stddev) of a characteristic for the single basins are based on the morphological input data at the $100 \times 100$ m$^2$ resolution.

**Table 1.** Basin properties and water balance characteristics of the seven major German river basins. The geographical location of the basins is depicted in Figure 1. Abbreviations: avg - average, std - standard deviation, min - minimum, max - maximum, P - precipitation, Q - streamflow, $E_a$ - evapotranspiration (P-Q), $E_p$ - potential evapotranspiration

| Major Basins | Basin Area [km$^2$] | Elevation [m] | | | | Land Cover [%] | | | Water Balance [mm a$^{-1}$] | | | Dryness Index [-] | Runoff Coeff. [-] |
|---|---|---|---|---|---|---|---|---|---|---|---|---|---|
| | | avg | std | min | max | forest | sealed | mixed | P | Q | $E_a$ | $E_p$/P | Q/P |
| Mulde | 6 200 | 386 | 201 | 75 | 1 212 | 26 | 10 | 64 | 798 | 344 | 454 | 0.88 | 0.43 |
| Ems | 8 400 | 60 | 36 | 10 | 383 | 13 | 8 | 79 | 802 | 312 | 490 | 0.89 | 0.39 |
| Neckar | 12 700 | 445 | 153 | 124 | 1 015 | 35 | 10 | 55 | 914 | 356 | 558 | 0.85 | 0.39 |
| Main | 23 700 | 356 | 113 | 93 | 1 044 | 39 | 6 | 55 | 793 | 247 | 546 | 0.97 | 0.31 |
| Saale | 24 800 | 287 | 162 | 56 | 1 139 | 23 | 8 | 69 | 645 | 161 | 484 | 1.13 | 0.25 |
| Weser | 37 700 | 223 | 165 | 8 | 1 116 | 34 | 7 | 59 | 781 | 276 | 505 | 0.91 | 0.35 |
| Danube | 47 500 | 558 | 170 | 302 | 2 329 | 32 | 6 | 62 | 948 | 469 | 479 | 0.80 | 0.49 |

**Table 2.** Evaluation of evapotranspiration $E_a$ and soil moisture $SM$ at seven eddy covariance stations. The evaluation is based on daily and monthly values for the available observation period. The location of the eddy stations is depicted in Figure 1. Abbreviations: RMSE - root mean squared error, $\rho$ - Pearson correlation coefficient, $E_a$ - evapotranspiration, SM - soil moisture

| ID | Station Name | Period | Land Cover | Monthly $E_a$ [mm mon$^{-1}$] | | [-] | Daily $E_a$ [mm d$^{-1}$] | | [-] | Daily SM [-] |
|---|---|---|---|---|---|---|---|---|---|---|
| | | | | RMSE | Bias | $\rho$ | RMSE | Bias | $\rho$ | $\rho$ |
| E1 | Gebesee | 2003-2008 | cropland | 19.14 | 0.61 | 0.85 | 1.01 | 0.02 | 0.67 | 0.62 |
| E2 | Hainich | 2000-2007 | DBF* | 11.72 | 6.99 | 0.95 | 0.62 | 0.23 | 0.87 | 0.68 |
| E3 | Mehrstedt | 2003-2006 | grasland | 12.44 | 5.78 | 0.94 | 0.74 | 0.18 | 0.79 | 0.80 |
| E4 | Wetzstein | 2004-2008 | ENF** | 9.86 | 1.58 | 0.96 | 0.73 | 0.05 | 0.84 | 0.80 |
| E5 | Grillenburg | 2004-2008 | grasland | 13.93 | -4.19 | 0.94 | 0.89 | -0.14 | 0.8 | 0.93 |
| E6 | Klingenberg | 2004-2008 | cropland | 15.39 | 9.38 | 0.93 | 0.86 | 0.31 | 0.77 | 0.53 |
| E7 | Tharandt | 1997-2008 | ENF** | 13.39 | 7.71 | 0.96 | 0.72 | 0.26 | 0.83 | 0.82 |

\* deciduous broadleaf forest, \*\* evergreen needleleaf forest