# Peer review of "A High-Resolution Dataset of Water Fluxes and States for Germany Accounting for Parametric Uncertainty"

_Hydrology and Earth System Sciences, 2016_

## Referee Comment (RC1) · Anonymous Referee #1 · 11 Nov 2016

Review of HESS-2016-443 "A High-Resolution Dataset of Water Fluxes and States for Germany Accounting for Parametric Uncertainty"

Overall:

This paper provides an excellent and useful product, and the complexities and methods used to derive this product. There are some minor organizational and grammatical errors, but overall I think this paper is a good contribution to broad-scale hydrologic modeling and analysis.

The paper could benefit from some additional attention to organization, specifically in the introductory sections. The authors limit themselves in the stated aim of the

paper in the introduction, and go on to state other aims later in the paper. The aim as stated in the introduction is "to derive a consistent set of national-scale hydrologic data for Germany at high spatial and temporal resolutions". If this was the extent of the paper, I would recommend that this be re-submitted as a methods paper; however, the authors go on to append additional aims/goals in the body of the paper which go beyond this, such as: Page 5, line 22: "to derived consistent model parameters to perform nationwide simulations of water fluxes and states." Page 7, line 8: "to analyze the temporal dynamics of soil moisture" Page 13, line 21: "spatio-temporal differences of uncertainties caused by the 100 ensemble parameter sets" As it is written, this paper reads as an aggregation of papers instead of one cohesive contribution. This could be easily fixed by restating the aim in the introduction to include all the parametric uncertainty analyses that are presented in the paper. Please gather and reassess the purpose and scope of the entire paper's contribution to the field of hydrologic science, and state this in the introduction. There are also several spots in the paper which could be improved by directly assessing the limitations of the data or analyses. An example of where this is done well in the paper is page 7, lines 7-8: "A direct comparison between observed and simulated soil moisture may therefore be misleading due to differences in spatial representativeness and sampling depth."

Specific comments:

Page 1, line 24: formatting of the citation does not match others. Page 2, line 6: state limitations of using observational data. Page 2, line 9: add "contiguous' or 'continental' United States. Page 2, line 13-14: grammar. ". . . who stated a need for higher-resolution spatial data and models. . . " Page 3, line 11: add 'entirely', "only catchment entirely covered by German territory" Page 3, line 17: grammar, "average discharge of the seven catchments ranges" Page 4, line 1-2: how does this assumption of static land cover before 1990 impact results? Page 4, line 2: what are your aggregation/resampling methods? Page 4, line 4: remove 'information' after 'gauging station' Page 4, line 7: what are the relative characteristics of the 222 catchments?

Size? Is there a map? Page 4, line 18: authors state that this spatial resolution is appropriate without additional reasoning or citation. Please provide one or both. Page 5, line 3: change 'precipitation' to 'rainfall'. Page 5, line 6: "On average, it is 1.8 m deep in Germany." Given the previous sentence this is confusing. What is the 'it' in this sentence? Page 5, line 11: since Germany is a part of Europe, "various river basins across Europe (including Germany), and the USA..." Page 5, line 14: remove 'the' before porosity. Page 5, line 22: this goal is not included in the introductory aim. Page 7, line 8-9: this aim is not included in the introduction. Page 8, line 1-2: mean and standard deviation symbols defined in line 1, just use symbols in following sentence. Page 8, line 25: choose Basins or Catchments and be consistent. I would recommend Basins, which would require going back and changing this throughout the paper. Page 8-9; this section could benefit from more organization, such as the use of a more explicit introductory paragraph. Presentation jumps from analysis of results overall to specific basins, which is difficult to follow. Page 9, line 26: remove 'is' at the end of the sentence. Page 10, lines 16-24: You discuss the energy- and water-limited conditions, which could also be added to the figure. It also appears that at this point the data switch from under representation to over representation (within the uncertainty bounds). Please discuss this. Page 11, line 5: your data groupings get confusing here. In the figure daily and monthly (I think?) data are grouped by color in seasons. In the table you report monthly and daily values. Be explicitly clear about what is being reported here. Maybe have a different symbol for monthly and daily data to differentiate (if they are both in the scatterplots... still not clear.). Page 11, line 6: "The results of the scatter plot... indicate..." this phrasing is awkward. Consider "The scatterplots shown in Figure 5 indicate..." Page 11, lines 17-23: add 'limitations' before Hargreaves-Samani approach to improve clarity of the paragraph. Page 11, line 24-page 12 line 2: does land cover type play a role in the ability to interpolate point to grid data? Are some land cover types likely to be more spatially heterogenous with respect to ET and soil moisture? Could this be incorporated into an uncertainty analysis? Page 12, line 11: these features (Central Uplands, Alps) are not on the map.

Tables:

Capitalize all headers, to be consistent (Table 1 no headers are capitalized, Table 2 some are, some aren't. Table 1. Header for Major German Basins Table 2. Describe RMSE, BIAS and in a footnote, and what "[-]" means in a footnote or caption. "Station-name" should be two words.

Figures:

Figure 1: what do colors represent? Please describe in figure caption. Figure 4: using the same color scale is a little misleading. Include the locations of the eddy covariance stations on the map. Figure 5: consider changing symbols so that this figure is readable in black and white. Figure 6: why are these four stations selected? Figure 7: the Central Uplands and Alps (referenced on page 12, line 11) are not explicitly shown on this map.

[Figure]

---

## Referee Comment (RC2) · Anonymous Referee #2 · 16 Nov 2016

The authors provide a description of a publicly available dataset that they have developed for Germany. Their product will be useful for the scientific community. Aside from a few problematic oversights, the paper is generally well-written, with appropriate figures and references. In my opinion the paper will be suitable for publication after a minor revision.

Major

A major oversight of this paper is the lack of referencing a relevant paper that provides a similar dataset, at least in scope. The dataset of Newman et al. (2015) is also a 100-sample ensemble and needs to be cited here. The similarities and differences of the authors dataset with that of Newman et al. (2015) should be noted.

[Figure]

It is surprising that ET would have less uncertainty than streamflow since the latter is a more direct measurement. The authors only evaluate ET at 7 locations, while discharge is evaluated at over 200. It seems inconsistent to suggest that uncertainty across these two observations could be readily compared. Additional discussion is warranted here, including the scale mismatch between a 4kmx4km grid cell and a point observation. Further, the authors should comment more directly on why they did not evaluate the spatial patterns of their model against remotely-sensed ET and consider doing this evaluation.

The validation watersheds range in size by nearly two orders of magnitude. If the model spatial resolution is the same for all, the authors should comment and hypothe-size whether they see higher model performance in larger basins—does performance increase monotonically with basin size?

In Figure 4, climatic regime does not appear to be a good predictor of model per-formance, with some of the highest NSE scores distributed throughout the range of conditions. The authors should comment on what, if anything, will best predict model performance, to guide a potential user of the dataset.

Minor

P1L24: Grammar: "have a footprints"

P1L24: "827 stations worldwide"—perhaps more apt to say "less than 1,000" loca-tions worldwide", since there are other observational sources beyond fluxnet.

P2L1: replace "reanalysis data" with "reanalysis products" and make this change else-where

P2L9: Maurer et al (2002) and Livneh et al. (2015) also cover a significant area in Canada (i.e. not just US, MX, and China).

P9: Here and elsewhere the use of the plural form of the word "performance" as "per-formances" is grammatically incorrect. Please correct this.

**References**

Newman, A. J., M. P. Clark, J. Craig, B. Nijssen, A. W. Wood, E. D. Gutmann, N. Mizukami, L. Brekke, and J. R. Arnold (2015). An observationally based gridded ensemble of precipitation and temperature data for the contiguous USA.ÂăJ. Hydrometeorology,Âădoi:10.1175/JHM-D-15-0026.1.

Livneh, B., Bohn, T. J., Pierce, D. W., Munoz-Arriola, F., Nijssen, B., Vose, R., & Brekke, L. (2015). A spatially comprehensive, hydrometeorological data set for Mexico, the US, and Southern Canada 1950–2013.ÂăScientific data,Âă2.

---

## Referee Comment (RC3) · Anonymous Referee #3 · 21 Nov 2016

In the submitted manuscript "A High-Resolution Dataset of Water Fluxes and States for Germany Accounting for Parametric Uncertainty" Zink et al. present a new approach to calibrate a distributed model (the mesoscale Hydrological Mode mHM) across all basins across Germany on a 4x4km$^2$ resolution. They use a 2 step calibration procedure, during which they firstly calibrate 7 major basins individually, and, secondly use a subset of the calibrated parameter samples with sufficient performance (NSE>0.65) at all 7 basins to apply them over the remaining catchments over Germany. Using split-sample tests and auxiliary information (AET, soil moisture, recharge) they evaluate the model and the combined parameter set concerning its general performance and uncertainty. Overall, the approach is well chosen and the provided results make sense.

However, the manuscript needs serious improvements before it can be considered for publication in HESS. Most of the points of criticism are related to the need for more rigorousness:

- The introduction is too short and does not provide a proper view on the research gaps of the approaches and methods applied in this study (for instance calibration and model evolution approaches). It appears to be series of vaguely related short paragraphs -> a more robust story line is needed.

- The methods are incomplete, partially referring to previous research, partially omitting parts of the analysis that later appears in the results section. On the other hand some information is irrelevant. Very important information, for instance introducing the model parameters that are calibrated, is completely missing. Up to the end of the manuscript it is not clear, which parameters were calibrated, which ranges were used and there was no discussion of their physical meaning.

- There is generally too little referencing of other studies. In particular in the Results and Discussion section, there are some interpretations that are hardly supported by the results and almost no comparison to the research of others.

- In general there is a lack of self-criticism. There are many obvious and hidden assumptions in the approach and the authors should spend significantly more effort discussing them.

For all these reasons, which are elaborated in more detail in the commented pdf, I recommend major revisions. I am convinced that the approach and the results are novel and reasonable but the authors have to show this in a rigorous scientific way.

Please also note the supplement to this comment:
http://www.hydrol-earth-syst-sci-discuss.net/hess-2016-443/hess-2016-443-RC3-supplement.pdf

[Figure]

**Supplement:**

[revised manuscript text omitted]

---

## Author Comment (AC1) · 19 Dec 2016

**1 Anonymous Referee #1**

**1.1 Overall:**

This paper provides an excellent and useful product, and the complexities and methods used to derive this product. There are some minor organizational and grammatical errors, but overall I think this paper is a good contribution to broad-scale hydrologic modeling and analysis.

Thank you for offering such encouraging and detailed suggestions for improving the structure of the paper. We appreciate your efforts to improve the structure and the clarity of the paper. We will revise the introduction as suggested by the reviewer. We added additional analyses for addressing the questions: 1) How the assumption of a static land cover before 1990 impacts the model results, and 2) What are the relative characteristics of the 222 catchments and how do they relate to model performance. In the following, we present the referee's comments as well as our point-by-point response to all of them.

The paper could benefit from some additional attention to organization, specifically in the introductory sections. The authors limit themselves in the stated aim of the paper in the introduction, and go on to state other aims later in the paper. The aim as stated in the introduction is to derive a consistent set of national-scale hydrologic data for Germany at high spatial and temporal resolutions. If this was the extent of the paper, I would recommend that this be re-submitted as a methods paper; however, the authors go on to append additional aims/goals in the body of the paper which go beyond this, such as: Page 5, line 22: to derived consistent model parameters to perform nationwide simulations of water fluxes and states. Page 7, line 8: to analyze the temporal dynamics of soil moisture Page 13, line 21: spatio-temporal differences of uncertainties caused by the 100 ensemble parameter sets As it is written, this paper reads as an aggregation of papers instead of one cohesive contribution. This could be easily fixed by restating the aim in the introduction to include all the parametric uncertainty analyses that are presented in the paper. Please gather and reassess the purpose and scope of the entire papers contribution to the field of hydrologic science, and state this in the introduction.

We thank the reviewer for his/her assessment of the manuscript. We will revise the manuscript based on your suggestion and reorganize the (methodological) aims in the introduction.

There are also several spots in the paper which could be improved by directly assessing the limitations of the data or analyses. An example of where this is done well in the paper is page 7, lines 7-8: A direct comparison between observed and simulated soil moisture may therefore be misleading due to differences in spatial representativeness and sampling depth.

We will address limitations of data and analyses where we identified them in the revised manuscript, e.g., the limitations of an observational driven, simulated hydrologic dataset.

Specific comments:

Page 1, line 24: Formatting of the citation does not match others.
We edited the format of the citation.
Page 2, line 6: State limitations of using observational data.

Thank you for pointing this out. We mention this limitations in the revised manuscript as "First, due to a limited amount of observed variables modeling approaches like the estimation of potential evapotranspiration have to be adopted to the available data. In consequence temperature based methods may be preferred to more physically based radiation approaches. Second, the interpolation of point observations induces uncertainties depending on the applied interpolation method. Further, small-scale, convective precipitation events may not be caught by gauging networks and lead to an underestimation in precipitation."

Page 2, line 9: add contiguous or continental United States.

Done.

Page 2, line 13-14: grammar. ... who stated a need for higher-resolution spatial data and models...

Thank you, done.

Page 3, line 11: add entirely, only catchment entirely covered by German territory

We changed that accordingly.

Page 3, line 17: grammar, average discharge of the seven catchments ranges

Thank you, done.

Page 4, line 1-2: How does this assumption of static land cover before 1990 impact results?

The impact of changing land cover is remarkably high near urban areas, since most of the changes there happened between 1950 and 2010. The effects of these changes are, however, low at the model resolution of $4 \times 4$km$^2$ as the table underneath shows. This table shows the differences between the hydrological state and fluxes between two scenarios. In the first scenario the land cover is fixed to the state of 1990. In the second scenario we fix the land cover to the conditions observed in 2006. The model time period is 1951 to 2010 and the domain is Germany. The comparison is based on daily values of the respective flux or state. Mean relative biases of less than 5% indicate that the assumption of static land cover has an impact on the modeled fluxes and states which is low compared to the effect of the parametric uncertainty. Changes apart from urbanization will have low effects on the modeled hydrological variables because mHM works with three land cover classes, i.e., sealed (mostly urban), forest and a mixed class. We restricted our study to only 3 land cover scenes because the well established CORINE data are only available for the years 1996, 2000, and 2006.

The table shows the mean and standard deviation between 2 land cover scenarios.

| variable | bias [mm d$^{-1}$] | rel. bias [%] | correlation [-] | RMSE [mm d$^{-1}$] |
|---|---|---|---|---|
| evapotranspiration | 0±0.02 | 0.22±1.46 | 1.0±0 | 0.01±0.03 |
| soil moisture | 0±0 | -0.02±0.67 | 1.0±0 | 0±0 |
| generated runoff | 0±0.02 | -0.57±4.91 | 1.0±0.03 | 0.03±0.07 |
| recharge | 0±0.01 | 0.43±2.79 | 1.0±0 | 0±0.01 |

Page 4, line 2: What are your aggregation/resampling methods?

We remapped the data using a nearest neighbor approach. We address this question in the revised manuscript.

Page 4, line 4: Remove information after gauging station

Done.

Page 4, line 7: What are the relative characteristics of the 222 catchments? Size? Is there a map?

The location of the catchments and their size can be retrieved from Figure 4 in the manuscript. For a better insight we relate basin specific characteristics to the model performance (see Figure 1 below). This figure is included and discussed in the revised manuscript. Furthermore, a table containing all the relevant information (location, mean elevation, mean slope, mean precipitation, etc.) will be published as supplement to this manuscript.

[Figure]

Figure 1: Relation between land surface and hydro-climatic conditions and model performance for the 222 river basins. The mean and standard deviation (stddev) of a characteristic for the single basins are based on the morphological input data at the $100 \times 100$ m$^2$ resolution. standard deviation.

Page 4, line 18: Authors state that this spatial resolution is appropriate without additional reasoning or citation. Please provide one or both.

We orientate the choice for a spatial resolution to the density of the precipitation stations. Two main arguments were considered: 1) The spatial resolution should be lower than the mean lowest distances between existing stations, 2) The chosen resolution shouldn't be so low that the interpolated meteorological variable is

mainly an artifact of the interpolation method (e.g., elevation driven external drift). Therefore, we argument that half of the minimum distances, i.e., 3 km, is a reasonable choice. Because of model specific reasons, we decided for the closest even number to 3 km, i.e., 4 km. We revised the manuscript to make this connection between spatial model resolution and average minimum distance between precipitation stations more clear.

Page 5, line 3: Change precipitation to rainfall.

Done.

Page 5, line 6: On average, it is 1.8 m deep in Germany. Given the previous sentence this is confusing. What is the it in this sentence?

We clarified this point in the revised manuscript.

Page 5, line 11: since Germany is a part of Europe, various river basins across Europe (including Germany), and the USA. . .

Done.

Page 5, line 14: remove the before porosity.

Done.

Page 5, line 22: This goal is not included in the introductory aim. Page 7, line 8-9: this aim is not included in the introduction.

Thank you. We address both points in revised the manuscript as stated above.

Page 8, line 1-2: mean and standard deviation symbols defined in line 1, just use symbols in following sentence.

Done.

Page 8, line 25: Choose basins or catchments and be consistent. I would recommend Basins, which would require going back and changing this throughout the paper.

We follow your recommendation and changed the manuscript accordingly.

Page 8-9:This section could benefit from more organization, such as the use of a more explicit introductory paragraph. Presentation jumps from analysis of results overall to specific basins, which is difficult to follow.

We will rewrite and reorganize this section.

Page 9, line 26: remove is at the end of the sentence.

Done.

Page 10, lines 16-24: You discuss the energy- and water-limited conditions, which could also be added to the figure. It also appears that at this point the data switch from under representation to over representation (within the uncertainty bounds). Please discuss this.

Since Figure 4 has already a high information content, we dare to add further information as, e.g., the separation to energy and water limited areas, by introducing another line or coloring. Nevertheless, we elaborated the caption of the figure to address this point. We discuss the issue of under- and overrepresentation in the text now. Thank you for pointing this out.

Page 11, line 5: Your data groupings get confusing here. In the figure daily and monthly (I think?) data are grouped by color in seasons. In the table you report monthly and daily values. Be explicitly clear about what is being reported here. Maybe have a different symbol for monthly and daily data to differentiate (if they are both in the scatterplots. . . still not clear.).

The scatterplot only shows daily data. We adopted the figure caption accordingly.

Page 11, line 6: "The results of the scatter plot... indicate..." this phrasing is awkward. Consider "The scatterplots shown in Figure 5 indicate..."

Thank you for your suggestion. We reformulated the sentence.

Page 11, lines 17-23: Add "limitations" before Hargreaves-Samani approach to improve clarity of the paragraph.

Done.

Page 11, line 24-page 12 line 2: Does land cover type play a role in the ability to interpolate point to grid data? Are some land cover types likely to be more spatially heterogenous with respect to ET and soil moisture? Could this be incorporated into an uncertainty analysis?

For the above mentioned analysis no spatial variabilities of morphological information are considered at all. The hydrological model is run on the point scale, i.e., a single grid cell ($100 \times 100$m$^2$), for this analysis. The relation of land cover and modeled evapotranspiration at the resolution of $4 \times 4$ km$^2$ is shown in Figure 8 and shortly discussed in section 4.5.

Page 12, line 11: These features (Central Uplands, Alps) are not on the map.

We changed the text in way that the location can be identified on the map in Figure 9.

Tables: Capitalize all headers, to be consistent (Table 1 no headers are capitalized, Table 2 some are, some arent.

We capitalized all table headers.

Table 1. Header for Major German Basins

We added the header in the revised manuscript.

Table 2. Describe RMSE, BIAS and in a footnote, and what [-] means in a footnote or caption. Station- name should be two words.

Thank you for pointing out that the abbreviations are not explained at all. We added the descriptions in the captions of both tables.

Figures:

Figure 1: What do colors represent? Please describe in figure caption.

The colors are only to better distinguish the different catchments. We added an explanation to the caption.

Figure 4: Using the same color scale is a little misleading. Include the locations of the eddy covariance stations on the map.

The location of the eddy covariance stations are depicted in Figure 1. We have chosen the same colors for the Budyko plot and the map plot to give the reader the opportunity to see wether model performances are clustering for a particular climatic regime and/or geographical location. For the maps on the right side, we are using the inverse color bar. We intend to ease up the comparison of catchments showing a relatively good performance (green) with relative low NSE ranges (green) and vice versa.

Figure 5: Consider changing symbols so that this figure is readable in black and white.

We changed the illustration of the different seasons to four different marker symbols.

Figure 6: Why are these four stations selected?

First, of all to represent the major mHM land cover classes (forest and mixed) are represented. And second, because they have at least continuous 3 year long time series without big data gaps. Further the four station are spread over the three regions where eddy covariance observations are available. We added this reasoning to the figure caption.

Figure 7: The Central Uplands and Alps (referenced on page 12, line 11) are not explicitly shown on this map.

We changed the text such that the location can be identified on the map in Figure 9.

---

## Author Comment (AC2) · 19 Dec 2016

**1 Anonymous Referee #2**

The authors provide a description of a publicly available dataset that they have developed for Germany. Their product will be useful for the scientific community. Aside from a few problematic oversights, the paper is generally well-written, with appropriate figures and references. In my opinion the paper will be suitable for publication after a minor revision.

Thank you for your helpful comments which are highly appreciated by us. The manuscript benefited from your suggested analyses and literature. We added the mentioned references to the revised manuscript and discuss them. Further, we address the questions: 1) Could the model performance of the 222 catchments be explained by any land surface or hydro-meteorological conditions?, and 2) How does the model estimate of ET compares with a remotely-sensed product? by additional analyses. In the following, we present the referee's comments as well as our point-by-point response to all of them.

**1.1 Major**

A major oversight of this paper is the lack of referencing a relevant paper that provides a similar dataset, at least in scope. The dataset of Newman et al. (2015) is also a 100-sample ensemble and needs to be cited here. The similarities and differences of the authors dataset with that of Newman et al. (2015) should be noted.

We discuss the difference in among these datasets in the introduction and the conclusions now. The mentioned references were added.

It is surprising that ET would have less uncertainty than streamflow since the latter is a more direct measurement. The authors only evaluate ET at 7 locations, while discharge is evaluated at over 200. It seems inconsistent to suggest that uncertainty across these two observations could be readily compared. Additional discussion is warranted here, including the scale mismatch between a $4 \times 4$ km$^2$ grid cell and a point observation.

The uncertainty of evapotranspiration and generated runoff is compared on the grid cell level, e.g., Figures 8 and 9. This comparison does not consider any observations. This analysis is based on the ensemble spread of the simulations at the $4 \times 4$ km$^2$ resolution. The model and model parameters are beforehand evaluated at point scale, i.e., $100 \times 100$ m$^2$, with observations at eddy covariance stations and at the $4 \times 4$ km$^2$ with discharge observations among others. For these evaluations we do not compare uncertainties of different variables as they would not be "readily comparable" because of the scale mismatch as you mentioned. So we fully agree with you in the argument that uncertainties from hydrologic variables at different resolutions are not comparable.

Further, the authors should comment more directly on why they did not evaluate the spatial patterns of their model against remotely-sensed ET and consider doing this evaluation.

Thank you for mentioning this point. We added a comparison of the ensemble mean of modeled evapotranspiration with MODIS evapotranspiration. We elaborated the results of this comparison in the revised manuscript and added Figure 1 to the manuscript).

[Figure]

Figure 1: Comparison of monthly estimates of evapotranspiration from mHM and MODIS in the period 2001-2010. The ensemble is represented by the ensemble mean of 100 evapotranspiration estimates. The comparison is based on three statistical assessments: A) relative bias, B) Pearson correlation coefficient, and C) root mean squared error (RMSE). The respective units are given in brackets.

The validation watersheds range in size by nearly two orders of magnitude. If the model spatial resolution is the same for all, the authors should comment and hypothesize whether they see higher model performance in larger basins. Does performance increase monotonically with basin size?

We comment on this issue in the manuscript as "However, a tendency to perform better in large catchments basins is observed." We also note that there is no clear (monotonic) relationship between basin area and NSE as can be observed from the Figure 2 below.

In Figure 4, climatic regime does not appear to be a good predictor of model performance, with some of the highest NSE scores distributed throughout the range of conditions. The authors should comment on what, if anything, will best predict model performance, to guide a potential user of the dataset.

Fortunately, we did not find any meteorological or morphological characteristics which explained why model performance is different for different catchments. This makes us confident that the retrieved parameter sets are representative for various climatic and physiographic conditions. We performed an analysis for identifying relations between land surface and hydro-climatic characteristics and model performance. Figure 3 was added in the revised manuscript and the findings are shortly discussed in section 4.2 of the revised manuscript.

[Figure]

Figure 2: Relation of model performance and catchment area for the 222 basins.

[Figure]

Figure 3: Relation between land surface and hydro-climatic conditions and model performance for the 222 river basins. The mean and standard deviation (stddev) of a characteristic for the single basins are based on the morphological input data at the $100{\times}100$ m$^2$ resolution. standard deviation.

**1.2 Minor**

P1L24: Grammar: have a footprints

Changed.

P1L24: 827 stations worldwide perhaps more apt to say "less than 1,000 locations worldwid", since there are other observational sources beyond fluxnet.

Changed.

P2L1: replace "reanalysis data" with "reanalysis products" and make this change elsewhere

Done.

P2L9: Maurer et al (2002) and Livneh et al. (2015) also cover a significant area in Canada (i.e. not just US, MX, and China).

Thanks for pointing out this fact. We changed the text "' accordingly.

P9: Here and elsewhere the use of the plural form of the word "performance" as "performances" is grammatically incorrect. Please correct this.

Changed.

References:

Newman, A. J., M. P. Clark, J. Craig, B. Nijssen, A. W. Wood, E. D. Gutmann, N. Mizukami, L. Brekke, and J. R. Arnold (2015). An observationally based gridded ensemble of precipitation and temperature data for the contiguous USA. J. Hydrometeorology, doi:10.1175/JHM-D-15-0026.1.

Livneh, B., Bohn, T. J., Pierce, D. W., Munoz-Arriola, F., Nijssen, B., Vose, R., & Brekke, L. (2015). A spatially comprehensive, hydrometeorological data set for Mexico, the US, and Southern Canada 19502013. Scientific data, 2.

---

## Author Comment (AC3) · 19 Dec 2016

**1 Anonymous Referee #3**

In the submitted manuscript "A High-Resolution Dataset of Water Fluxes and States for Germany Accounting for Parametric Uncertainty" Zink et al. present a new approach to calibrate a distributed model (the mesoscale Hydrological Mode mHM) across all basins across Germany on a $4 \times 4$ km$^2$ resolution. They use a 2 step calibration procedure, during which they firstly calibrate 7 major basins individually, and, secondly use a subset of the calibrated parameter samples with sufficient performance (NSE$\geq$0.65) at all 7 basins to apply them over the remaining catchments over Germany. Using split-sample tests and auxiliary information (AET, soil moisture, recharge) they evaluate the model and the combined parameter set concerning its general performance and uncertainty. Overall, the approach is well chosen and the provided results make sense.

However, the manuscript needs serious improvements before it can be considered for publication in HESS. Most of the points of criticism are related to the need for more rigorousness.

We would like to thank the reviewer for his/her valuable comments. We highly appreciate them. We think the manuscript improved significantly by addressing these comments. Based on the comments of the reviewer we revised the introduction and the results and discussion section. We discuss additional references and strengthened the rigorousness of the manuscript and the scientific analyses therein. In the following, we present the referee's comments as well as our point-by-point response to all of them.

- The introduction is too short and does not provide a proper view on the research gaps of the approaches and methods applied in this study (for instance calibration and model evolution approaches). It appears to be series of vaguely related short paragraphs - a more robust story line is needed.

  We will add a paragraph discussing the calibration of hydrologic models for large spatial domains to the introduction.

- The methods are incomplete, partially referring to previous research, partially omitting parts of the analysis that later appears in the results section. On the other hand some information is irrelevant. Very important information, for instance introducing the model parameters that are calibrated, is completely missing. Up to the end of the manuscript it is not clear, which parameters were calibrated, which ranges were used and there was no discussion of their physical meaning.

  For giving deeper insight to the model parameterization we rewrote the model description part (section 3.1) which made it hopefully better understandable. Further, we added tables of the effective model parameters in the revised manuscript. A deeper insight to the model and model parameterization is however out of the scope of this study. We refer to Samaniego et al. 2010 and Kumar et al. 2103 (also mentioned in the manuscript) for further details.

  Samaniego, L., Kumar, R., & Attinger, S. (2010). Multiscale parameter regionalization of a grid-based hydrologic model at the mesoscale. Water Resources Research, 46(5), W05523.

  Kumar, R., Samaniego, L., & Attinger, S. (2013). Implications of distributed hydrologic model parameterization on water fluxes at multiple scales and locations. Water Resources Research, 49(1), 360379.

- There is generally too little referencing of other studies. In particular in the Results and Discussion section, there are some interpretations that are hardly supported by the results and almost no comparison to the research of others.
  We will address the need for more references in the discussion of the results by adding comparisons to similar studies where appropriate, e.g., to Newman et al. 2015.
  Newman, A. J., Clark, M. P., Craig, J., Nijssen, B., Wood, A., Gutmann, E., Arnold, J. R. (2015). Gridded Ensemble Precipitation and Temperature Estimates for the Contiguous United States. Journal of Hydrometeorology, 16(6), 24812500. doi: 10.1175/JHM-D-15-0026.1

- In general there is a lack of self-criticism. There are many obvious and hidden assumptions in the approach and the authors should spend significantly more effort discussing them.
  We added discussions of limitations of chosen approaches and assumptions at places in the manuscript which could be identified by us, and which were pointed out by you (see point-by-point answers).

For all these reasons, which are elaborated in more detail in the commented pdf, I recommend major revisions. I am convinced that the approach and the results are novel and reasonable but the authors have to show this in a rigorous scientific way.

Introduction: more structure needed, storyline incomplete too general, mixed up with results
We reorganized and rewrote major parts of the introduction.
P1L8: please explain acronym
Done.
P2L1: reanalysis data: What type of data?
We added some examples of potential reanalysis data.
P2L5: observational data: Please be more specific on Scale and type of data
Thanks for the comment. We specified what we mean with observational data in the revised manuscript.
P2L17-21: You mention observational uncertainty and then you decide to only consider parameter uncertainty. Please establish link between these different types of uncertainty.
In this paragraph of the introduction we gave an overview on all possible sources of predictive uncertainty in hydrologic modeling but surely we can not pursue all aspects of uncertainty within a single paper. Therefore, we aim to nalyze other sources of uncertainty in separate studies, e.g., Baroni et al. 2016. A discussion about the links between the different sources of uncertainty is added to the revised manuscript.
Baroni, G., Zink, M., Kumar, R., Samaniego, L., and Attinger, S.: On the effect of the uncertainty in soil properties on the simulated hydrological state and fluxes at different spatio-temporal scales, Hydrol. Earth Syst. Sci. Discuss., doi:10.5194/hess-2016-657, in review, 2016.

P2L27-28: This is already results - don't mention here
We deleted the respective sentence.
P3L26: hydrogeological vector map: please clarify: is this a hydro geological map?
Yes it is a hydrogeological map. We have provided a reference to this map for further details. We also revised wording to "hydrogeological map".
P4L12: On average? What does this mean when referring to the number of stations?
The number of meteorological stations is varying over time. New stations are established while others are disassembled. We provide the average number of stations of the modeled time period of 1951-2010.
P4L19: Why are you not using a more physically based method like Penman Monteith?
We use the well established Hargreaves-Samani approach in this study because it has the best support with observational data. As mentioned in the paper we use about 570 climate stations over Germany for providing input to the Hargreaves-Samani method. In contrast radiation observations are sparsly conducted within Germany. Right now approximately 80 global radiation measurement stations exist in Germany and still longwave radiation information are missing. Therefore, we can not estimate PET based on the Penman-Monteith approach. Moreover, several studies showed that PET estimates of regionalized Hargreaves-Samani approaches are close to those of Penman-Montheith estimates. Herein we are using a regionalized Hargreaves-Samani approach which is based on the aspect of the respective grid cell.
further reading:

Almorox, J., Quej, V. H., & Mart, P. (2015). Global performance ranking of temperature-based approaches for evapotranspiration estimation considering Köppen climate classes. Journal of Hydrology, 528, 514522. doi:2015.06.057

Droogers, P., & Allen, R. G. (2002). Estimating reference evapotranspiration under inaccurate data conditions. Irrigation and Drainage Systems, 16(1), 3345. doi:10.1023/A:1015508322413

Temesgen, B., Eching, S., Davidoff, B., & Frame, K. (2005). Comparison of Some Reference Evapotranspiration Equations for California. Journal of Irrigation and Drainage Engineering, 131(1), 7384. doi:(ASCE)0733-9437(2005)131:1(73)

P4L23: REGNIE: Why didn't you use this data as direct input for the model?
First, The German Meteorological Service was working at the development of the REGNIE data set in parallel to us. So after we finished the establishment of our interpolation routines in 2011 the REGNIE product was released. Second, we intended to use daily updated station data from the German Meteorological Service for running hydrological simulations on an operational basis. We could realize this aim in 2014 (www.ufz.de/droughtmonitor). And third, we publish our precipitation data set herein to address the need of investigating and analyzing input data uncertainties. Since both interpolation approaches are based on different methodologies we consider the publication of an alternative gridded precipitation product as added value for future research activities.
The mesoscale Hydrologic Model mHM: parameter estimation not clear
We now elaborate more on the estimation of parameters within mHM in the revised manuscript. A detailed description of the Multiscale Parameter Regionalization technique is out of the scope of this study since it was already published

in Samaniego et al. 2010 and Kumar et al. 2013. We refer to those papers for getting deeper insight to the parameterization of mHM.

P5L19-20 Is the sub grid variability also up scaled by distribution functions or is it finally One effective value derived by sub grid information?

The effective parameter is an effective value which was derived by sub grid information. We clarified this in the manuscript.

P5L22: How many calibration parameters do you have?

We use 67 *global* or *transfer* parameters which were calibrated. We mention this fact in the revised manuscript. We add an overview of these parameters and their ranges to the supplementary material.

P5L25: It is not clear how the different parameter sets derived from the 7 basins are put together to be used at the remaining basins.

We transfer the *global* parameters which were inferred by calibration from one catchment to another (receiver) basins. mHM allows for this flexibility because the *global* parameters are time-invariant and location-independent. These parameters are then used for the hydrologic simulation in each of the receiver catchments.

P5L26: Mention studies that used similar approaches for parameter estimation and model evaluation such as

Choi, H. T. and Beven, K.: Multi-period and multi-criteria model conditioning to reduce prediction uncertainty in an application of TOPMODEL within the GLUE framework, J. Hydrol., 332(34), 316336, doi:10.1016/j.jhydrol.2006.07.012, 2007.

Hartmann, A., Gleeson, T., Rosolem, R., Pianosi, F., Wada, Y. and Wagener, T.: A large-scale simulation model to assess karstic groundwater recharge over Europe and the Mediterranean, Geosci. Model Dev., 8(6), 17291746, doi:10.5194/gmd-8-1729-2015, 2015. and there are surely more if you take a closer look

We will discuss other approaches, e.g., the above mentioned approaches, for model evaluation and parameter estimation in the revised manuscript.

P6L13: Is this number large enough to find the best parameter sets?

As the results in Figure 1 shows this number iterations is sufficient to obtain reasonable performances. We have to admit the dynamically dimensioned search algorithm will not find optimal parameter values. This algorithm is design to find sufficient objective function values in a reasonable amount of time. Consequently another algorithm, e.g., the Shuffled Complex Evolution algorithm, needs to be applied for identifying the optimum of the objective function. For the herein proposed purpose the choice for DDS is reasonable because the aim is to identify reasonable parameter sets, rather than the best ones, for a set of 7 big catchments in a reasonable amount of time. The results of the model calibration are shown in Figure 2 as white boxes in the upper left corner. With exception of the Saale river basin all catchments reveal sufficient discharge estimations (median NSE$\geq$0.85, overall mean NSE=0.89).

P7L1: What justifies this? Other studies discarded time periods during the energy balance which could not be closed (e.g. Miralles, D. G., De Jeu, R. A. M., Gash, J. H., Holmes, T. R. H. and Dolman, A. J.: Magnitude and variability of land evaporation and its components at the global scale, Hydrol. Earth Syst. Sci., 15(3), 967981, doi:10.5194/hess-15-967-2011, 2011.")

The energy balance is not closed on the majority of the eddy flux towers worldwide due to a variety of reasons (e.g., Stoy et al. 2007, Foken 2008, Leuning

2012). There is an extensive literature on how to correct the observed fluxes (e.g., Twine et al. 2000, Wilson et al. 2002, Boldocchi 2003, Stoy et al. 2007, Allen 2008, Hendricks Franssen et al. 2010, Mauder et al. 2010, 2013, Foken et al. 2011, Kessomkiat et al. 2013, Charuchittipan et al. 2014, Ingwersen et al. 2015) ranging from correcting mostly latent heat to correcting mostly sensible heat. Two prominent arguments, which show immediately why latent heat should be corrected as well, are 1. meso-scale circulations that remove energy horizontally, i.e., in a movement perpendicular to the tower observations (e.g., Stoy et al. 2007) and 2. dampening of the water vapour signal in the tubing of the so-called closed path analysers and hence loss of high-frequency contributions especially for latent heat (e.g., Leuning 2012). We use a conservative correction, which is similar to preserving the observed Bowen ratio.
further reading:

Allen, R. G. (2008), Quality assessment of weather data and micrometeological flux - Impacts on evapotranspiration calculation, Journal of Agricultural Meteorology, 64(4), 191204.

Baldocchi, D. D. (2003), Assessing the eddy covariance technique for evaluating carbon dioxide exchange rates of ecosystems: past, present and future, Global Change Biology, 9(4), 479492.

Charuchittipan, D., W. Babel, M. Mauder, J.-P. Leps, and T. Foken (2014), Extension of the averaging time in Eddy-covariance measurements and its effect on the energy balance closure, Boundary-Layer Meteorology, 152(3), 303327, doi:10.1007/s10546-014-9922-6.

Foken, T., M. Aubinet, J. J. Finnigan, M. Y. Leclerc, M. Mauder, and K. T. Paw U (2011), Results of a panel discussion about the energy balance closure correction for trace gases, Bulletin of the American Meteorological Society, 92(4), ES13ES18, doi:10.1175/2011BAMS3130.1.

Hendricks Franssen, H. J., R. Stckli, I. Lehner, E. Rotenberg, and S. I. Seneviratne (2010), Energy balance closure of eddy-covariance data: A multisite analysis for European FLUXNET stations, Agricultural and Forest Meteorology, 150(12), 15531567, doi:10.1016/j.agrformet.2010.08.005.

Ingwersen, J., K. Imukova, P. Hgy, and T. Streck (2015), On the use of the post-closure methods uncertainty band to evaluate the performance of land surface models against eddy covariance flux data, Biogeosciences, 12, 23112326, doi:10.5194/bg-12-2311-2015.

Kessomkiat, W., H.-J. H. Franssen, A. Graf, and H. Vereecken (2013), Estimating random errors of eddy covariance data: An extended two-tower approach, Agricultural and Forest Meteorology, 171-172, 203219, doi: 10.1016/j.agrformet.2012.11.019.

Mauder, M., R. L. Desjardins, E. Pattey, and D. Worth (2010), An attempt to close the daytime surface energy balance using spatially-averaged flux measurements, Boundary-Layer Meteorology, 136(2), 175191, doi:10.1007/s10546-010-9497-9.

Mauder, M., M. Cuntz, C. Dre, A. Graf, C. Rebmann, H.-P. Schmid, M. Schmidt, and R. Steinbrecher (2013), A strategy for quality and uncertainty assessment of long-term eddy-covariance measurements, Agricultural and Forest Meteorology, 169, 122135, doi:10.1016/j.agrformet.2012.09.006.

Stoy, P. C., S. Palmroth, A. C. Oishi, M. B. S. Siqueira, J.-Y. Juang, K. A. Novick, E. J. Ward, G. G. Katul, and R. Oren (2007), Are ecosystem carbon inputs and outputs coupled at short time scales? A case study from

adjacent pine and hardwood forests using impulse-response analysis, Plant, Cell & Environment, 30(6), 700710, doi:10.1111/j.1365-3040.2007.01655.x.

Twine, T. E., W. P. Kustas, J. M. Norman, D. R. Cook, P. R. Houser, T. P. Meyers, J. H. Prueger, P. J. Starks, and M. L. Wesely (2000), Correcting eddy-covariance flux underestimates over a grassland, Agricultural and Forest Meteorology, 103(3), 279300.

Wilson, K. et al. (2002), Energy balance closure at FLUXNET sites, Agricultural and Forest Meteorology, 113(1-4), 223243.

P8L20: These are no results - delete or move to methods section

Thank you for pointing this out. We moved the sentence to the methods section.

P8L26: Fig 2 and text do not fit well together..,

We will rewrite the corresponding section in the revised manuscript.

P8l29: How is it possible that some of the grey whiskers fall below 0.65?

The parameter selection procedure is applied to model performances on daily basis in the validation period (see section 3.2). Thus, the grey boxes in the upper right panel of Figure 2 show the resulting performances after parameter selection. All of the whiskers are exceeding an NSE of 0.65.

P9L1: There are large drops for Mulde, Neckar and Danube - how can the average drop b. only 6%?

For clarification we provide the numbers below. As can be seen the average drop is 5.64%.

| medians | Mulde | Ems | Neckar | Saale | Main | Weser | Danube | average |
|---|---|---|---|---|---|---|---|---|
| NSE on-site | 0.80 | 0.82 | 0.90 | 0.69 | 0.92 | 0.91 | 0.84 | |
| NSE ensemble | 0.69 | 0.78 | 0.79 | 0.72 | 0.86 | 0.91 | 0.83 | |
| difference | -0.12 | -0.04 | -0.10 | 0.03 | -0.06 | 0.00 | -0.01 | -0.04 |
| normalization *100 [%] | -16.76 | -5.44 | -13.03 | 4.44 | -7.14 | 0.00 | -1.57 | -5.64 |

P9L3: What about Danube and Main? For those two the ranges change significantly

We revised the text to reflect your comment.

P9L9: Compensate for errors in the model structure: Provide some references to such cases.

We added a reference which is analyzing this problem (Clark and Vrugt 2006 ).

Clark, M. P., & Vrugt, J. a. (2006). Unraveling uncertainties in hydrologic model calibration: Addressing the problem of compensatory parameters. Geophysical Research Letters, 33(6), L06406. doi:10.1029/2005GL025604

P10L16: Should be mentioned in discussion

The manuscript does not have a separate discussion section. Therefore we included those discussion in the "Results and Discussion" section.

P10L24: 0.1: of what? NSE?

Yes. We revised the manuscript.

P11L17-20: So why not using a more physical approach?

Because an approach like Penman-Monteith (PM) is based on observations which are usually sparsely available as we elaborated beforehand. Thus, estimating evapotranspiration based on PM would imply to apply reanalysis products which introduce another degree of uncertainties because these data are partly relying on model estimations. The intention of this study was to use observational forcing data.

P12L27-P13L3: Doesn't this rather belong to the study site description?

These analyses are based on the gridded precipitation and potential evapotranspiration product which were develop in this study. For this reason we think these analyses are appropriately placed in the result part.

P13L6-9: Can you quantify the strength of the relation between AET/Q uncertainty and its explanatory variables?

Thank you for this comment. We quantified the strength of the connections between uncertainty patterns of the evapotranspiration and generated runoff with porosity and dryness index using the Spearman rank correlation (see table below). We elaborated on these results in the revised manuscript.

| Spearman rank corr. | Porosity | Dryness index |
| --- | --- | --- |
| Evapotranspiration uncertainty | 0.58 | 0.28 |
| Generated runoff uncertainty | 0.32 | 0.92 |

P13L9-10: You cannot state this without a proper sensitivity analysis

We reformulated this sentence.

P13L12-13: see my previous comment and provide numbers

Done.

P13L15-17: See previous comments. Right now, the data does not support such strong statements

We removed this sentence.

P13L30-31: Why? Please clarify.

We elaborated on that in the revised manuscript.

P14L9-14: Without mentioning or explaining the model parameters and visualizing that relationship between the snow and the soil parameters this statement is not supported be the analysis.

We will revise the manuscript accordingly and delete statements which are not supported by the data and parameters.

P14L25-26: You cannot state this without discussing actual values of the parameters. An acceptable NSE does not mean that the related model parameters are sensitive.

We kindly disagree with the interpretation of the reviewer. We are not aiming identifying parameter sensitivities in this study. Our aim is to find reasonable parameter sets on the basis of observed discharge. As we demonstrate within this study the chosen method can yield reasonably good model performance for discharge evaluation and is able to capture the spatio-temporal variability of $ET$ data.

Figure 2: Shouldn't this be filled white?

These boxes are filled white. The impression that they are grey may arise because of the narrow boxes. We assume that potential readers of the plot will assess the rationale behind the plot and interpret it in the right way as you did.

Figure 4: panel D?

Thank you. Changed.

Figure 6: observations hardly visible - please improve

We improved the plot.

Figure 9: The information on the range of uncertainties is provided by the grey area enveloping the median. I don't think the normalized range adds significantly more information to that - delete?

We argue that the normalization of the ranges is needed for the comparison of uncertainties among the hydrologic variables. Also interpreting the uncertainty behavior through the course of a year is more difficult without proper normalization. The uncertainty in evapotranspiration, for example, does not significantly change over the course of a year. Such a behavior would be difficult to observe without the normalized ranges. For that reason we prefer to stay with the figure as it is.